# Distinguishing ice-rich and ice-poor permafrost to map ground temperatures and ground ice occurrence in the Swiss Alps

Robert Kenner[1], Jeannette Noetzli[1], Martin Hoelzle[2], Hugo Raetzo[3], Marcia Phillips[1]

[1] WSL Institute for Snow and Avalanche Research SLF
[2] University of Fribourg, Department of Geosciences
[3] Federal Office for the Environment FOEN

*Correspondence to*: Robert Kenner (kenner@slf.ch)

**Abstract.** Mountain permafrost is invisible and mapping it is still a challenge. Available permafrost distribution maps often overestimate the permafrost extent and include large permafrost-free areas in their permafrost zonation. In addition, the representation of the lower belt of permafrost consisting of ice-rich features such as rock glaciers or ice-rich talus slopes can be challenging. These problems are caused by considerable differences in genesis and thermal characteristics between ice-poor permafrost occurring for example in rock walls, and ice-rich permafrost. While ice-poor permafrost shows a strong correlation of ground temperature with elevation and potential incoming solar radiation, ice-rich ground does not show such a correlation. Instead, the distribution of ice-rich ground is controlled by gravitational processes such as the relocation of ground ice by permafrost creep or by ground ice genesis from avalanche deposits or glacierets covered with talus.

We therefore developed a mapping method which distinguishes between ice-poor and ice-rich permafrost and tested it for the entire Swiss Alps. For ice-poor ground we found a linear regression formula based on elevation and potential incoming solar radiation which predicts borehole ground temperatures at multiple depths with an accuracy higher than 0.6° C. The zone of ice-rich permafrost was defined by modelling the deposition zones of alpine mass wasting processes. This dual approach allows the cartographic representation of permafrost-free belts, which are bounded above and below by permafrost. This enables a high quality of permafrost modelling, as is shown by the validation of our map. The dominating influence of the two rather simple connected factors elevation (as a proxy for mean annual air temperature) and solar radiation on the distribution of ice-poor permafrost is significant for permafrost modelling in different climate conditions and –regions. Indicating temperatures of ice-poor permafrost and distinguishing between ice-poor and ice-rich permafrost on a national permafrost map provides new information for users.

## 1 Introduction

Maps of potential permafrost distribution are useful products applied in different fields of practice and research because permafrost is an invisible subsurface phenomenon. Such maps are used to plan construction work in alpine terrain, to evaluate local slope instability or to estimate large-scale permafrost occurrence for scientific purposes. Mapping permafrost in the highly variable alpine landscape is however challenging, particularly on a global scale where ground temperature data or climate and terrain datasets are rare (Fiddes et al., 2015; Gruber, 2012). Developing a method appropriate to model mountain permafrost therefore requires test areas with a dense set of reference and validation data, as well as highly resolved digital terrain models. The Swiss Alps are an ideal test site, as various research activities during the last decades provide a ground temperature dataset, which is largely included in the Swiss permafrost monitoring network PERMOS (2016). Consequently, many authors have used the Swiss dataset to calibrate or validate their permafrost distribution model (Böckli et al., 2012; Deluigi et al., 2017; Fiddes et al., 2015; Gruber et al., 2006; Gruber and Hoelzle, 2001; Haeberli et al., 1996; Hoelzle et al., 2001; Keller, 1992; Keller et al., 1998).

The core of these models is a more or less simplified surface energy balance. Typically, mean annual air temperature (MAAT), represented by elevation and potential incoming solar radiation (PISR) are basic parameters (Hoelzle and Haeberli, 1995). Some authors only use MAAT (Azócar Sandoval et al., 2017) or freezing degree-days and snow cover (Gisnås et al., 2017; Ishikawa, 2003) as external forcing parameters. Fiddes et al. (2015) also consider precipitation and in particular snow cover, wind, humidity and a complete surface radiation balance in a purely physics-based method. Most other studies however used empirical-statistical approaches to define a permafrost likelihood or index based on the energy balance results and in dependency of landforms, surface coverage, vegetation or topographic characteristics such as slope or curvature (Böckli et al., 2012; Deluigi et al., 2017; Gruber, 2012; Hoelzle et al., 1993).

However, the actual distribution of mountain permafrost includes phenomena which cannot be sufficiently explained using surface energy balances. This mainly concerns the existence of excess ground ice at the base of talus slopes or in rock glaciers (Haeberli, 1975), which often occur hundreds of metres below the zone of continuous permafrost and are surrounded by permafrost-free ground. This type of permafrost, hitherto referred to as ice-rich permafrost, sometimes exists at locations with higher annual surface heat fluxes than in the surrounding permafrost-free areas (Lerjen et al., 2003; Scapozza et al., 2011). The permafrost-free belts between the ice-rich permafrost at lower elevations and permafrost with lower ice contents at higher elevations are not reproduced in the existing large scale mountain permafrost maps, as was highlighted by Lerjen et al. (2003) and Scapozza et al. (2011). This is because thermally defined maps have no information on ground ice content.

Ice-rich permafrost can persist under warmer climate conditions than ice-poor permafrost due to the high heat capacity of ice (Scherler et al., 2013). Due to latent heat effects, active layer thickness deepening was less pronounced in ice-rich permafrost than at ice-poor monitoring sites in the Swiss Alps during the last two decades (PERMOS, 2016). However, if active layer thickening did occur, it was reversible in ice-poor permafrost (Hilbich et al., 2008; Krautblatter, 2009; Marmy et al., 2013), but irreversible in ice-rich permafrost due to the melt of considerable amounts of ground ice (Zenklusen Mutter and Phillips, 2012). This highlights ground ice as a requirement for the existence of permafrost at ice-rich, low-elevation sites. It is therefore a logical step to

consider the ice content when mapping permafrost distribution, just as it is done for physics-based permafrost modelling (Hipp et al., 2012; Pruessner et al., 2018; Staub et al., 2015).

The differentiation between ice-rich and ice-poor permafrost was performed indirectly in earlier studies by including concave footslope positions in permafrost distribution models (Ebohon and Schrott, 2008; Keller, 1992). The permafrost and ground ice map (PGIM) presented here aims to reproduce the elevational permafrost gap by providing a better delimitation of the two main types of permafrost in alpine terrain. We consider the distribution of the continuous zone of ice-poor permafrost (permafrost without excess ice) as being mainly controlled by the surface energy fluxes. While negative temperatures allow small amounts of persistent ground ice in ice-poor permafrost, we assume the opposite for the ice-rich permafrost: Here, the ground ice enables the existence of permafrost, decoupled from current atmospheric conditions and often protected by coarse talus at the surface (Scherler et al., 2014; Schneider et al., 2012). The origin of this ground ice can be syngenetic due to the burial of snow and surface ice by rock debris (Haeberli and Vonder Mühll, 1996; Kenner, 2018) or epigenetic if originating from colder climate periods and displaced by long term rock glacier creep (Haeberli, 2000). To include both ice-poor and ice-rich permafrost in our map, we consider the surface energy balance in our model, which is decisive for the distribution of ice-poor permafrost. We also consider ground ice formation and relocation due to mass wasting processes, which control the distribution of ice-rich permafrost.

## 2 Methods

The permafrost and ground ice map (PGIM) of Switzerland distinguishes two alpine permafrost zones: zone 1 indicates modelled ground temperatures and is based on the parameters elevation and PISR. Zone 2 indicates areas outside of zone 1 which might be categorized as permafrost due to the existence of excess ground ice. The modelling approach for zone 2 differs fundamentally from that of zone 1: whereas zone 1 considers thermal conditions, the potential existence of ground ice is considered in zone 2; either due to superimposed rock fall and snow avalanche deposits or due to the gravity-driven relocation of excess ground ice.

### 2. 1 Mapping approach for zone 1

Zone 1 of the PGIM was derived from modelled ground temperatures. The ground temperatures were calculated based on a multiple linear regression analysis using the explanatory variables PISR and elevation (as a proxy for mean annual air temperature). These are the two most important parameters for the surface energy balance (Hoelzle et al., 2001) and are used in almost all permafrost distribution models. Ground temperatures measured in 15 reference boreholes were used as predictor variables. These boreholes were chosen from areas without ice-rich permafrost in Switzerland and Italy, close to the Swiss border (upper 15 sites in table 1). Temperature is measured by thermistors in the boreholes at multiple depths between 15 and 100 m with a sub-daily temporal resolution. The thermistors commonly have a measurement accuracy of around 0.1°C or better, and the types of thermistor and data loggers are specified in PERMOS (2016).

The basic concept was to attribute a PISR value, an elevation value and a mean annual ground temperature (MAGT) to each of the 212 thermistors. Based on this dataset, the regression parameters a, b and c in formula 1

were determined and later used in formula 4 to calculate the ground temperatures in zone 1. The four detailed work steps involved are explained below.

$$(1) \qquad MAGT = a + b \cdot R + c \cdot E$$

Where:

$MAGT$   is the mean annual ground temperature measured by each individual borehole thermistor

$R$         is the solar radiation value for each individual thermistor

$E$         is the elevation value for each individual thermistor

**Step 1: calculating solar radiation values for the ground surface**

PISR at the ground surface around each borehole was calculated using the ESRI tool "Area solar radiation". The
processing was based on a digital elevation model with 2 m resolution (swisstopo swissALTI3D). The input parameter transmissivity was set at 0.4, and the diffuse proportion at 0.5, which corresponds to values recommended for moist temperate climates by the software developer. Most of the alpine ground surface is snow-covered during large parts of the year and receives no insolation during that time. However, steep areas such as rock walls remain snow-free for the entire year (Magnin et al., 2015). To consider the snow cover in
slopes below 40°, we only used PISR values calculated for the generally snow-free period June to November (formula 2.1).

For slopes exceeding 40° we additionally included the winter solar radiation. This slope threshold lies within the zone in which winter snow cover clearly decreases (Pogliotti, 2011). Especially in sunny slopes, steeper than 40° C, winter insolation causes a positive feedback: Firstly, it causes snow removal due to melt or the triggering of
wet snow slides and subsequently an effective heating of the bare ground above the mean air temperatures (Haberkorn et al., 2015a). This in turn accelerates melt of the remaining snow. In steep, shady slopes however, winter insolation is often not strong enough to remove snow. In extremely steep parts in which snow cannot accumulate, long-wave radiation emission largely compensates the small amounts of incoming solar radiation in north facing slopes. This causes rock surface temperatures close to the air temperatures (Haberkorn et al., 2015a)

Our simplified model does not consider the emission of long-wave radiation and any additional winter insolation leads to a warming of the ground on an annual basis. As described above, this might be correct for southern slopes but not for northern ones. To overcome this weakness, the winter insolation (December to May) which affects the steep terrain parts was multiplied with an aspect-dependent factor. This factor ranges between 0 for the azimuth North (no effect of winter insolation due to similar strong long-wave emission) and 1 for the
azimuth South (strongest effect of winter insolation due to snow removal). The winter solar radiation was then added to the summer solar radiation values and applied to slopes steeper than 40° (formula 2.2).

For slopes < 40°:         (2.1)     $r = PISR_{June-Nov}$

For slopes > 40°:         (2.2)     $r = PISR_{June-Nov} + PISR_{Dec-May} \cdot A$

Where:

$r$         is the solar radiation value at a single surface point

PISR    is the potential incoming solar radiation

A        is an aspect factor ranging from 0 (N) to 0.5 (E/W) and 1 (S)

**Step 2: attributing solar radiation and elevation values to each borehole thermistor**

To attribute PISR and elevation to a thermistor we created a point cloud with 2 m resolution, representing the ground surface around each borehole. Every point contained information on its elevation and PISR. Radiation and elevation values for all surface points surrounding a thermistor influence its MAGT. To aggregate all these values into one radiation and elevation value representative for a thermistor, a spatial average was calculated (formula 3, as for elevation). The closer a surface point is to a thermistor, the stronger its influence. This was considered by an inverse distance weighting (factor d in formula 3). The larger the distance between a thermistor and a surface point, the higher the number of points lying within this distance. This increases the weight of distant surface areas when calculating a spatial average. To avoid this we categorized all points into distance classes with a 1 m increment, and included a second weighting factor considering the number of points within one distance class (factor k in formula 3). The maximal distance considered was 5 times the minimal distance of the thermistor to the ground surface. This factor was parameterised empirically by minimalizing the sum of residuals between measured and modelled ground temperatures.

$$(3) \qquad R = \frac{\sum_{i=n}^{i=1} d_i \cdot r_i \cdot k_i}{n}$$

Where:

$R$        is the solar radiation value defined for each individual borehole thermistor

$n$        is the number of distance classes

$d$        is a weighting factor which considers the distance between a surface point and the thermistor (inverse distance weighting)

$k$        is a weighting factor which considers the number of surface points within one distance class

$r$        is the solar radiation value of a single surface point

**Step 3: Setting up the regression model**

We analysed the dataset of step 2 in a multiple linear regression corresponding to formula 1. Naturally, the measured MAGT of a single thermistor deviates from the regression line towards warmer or colder conditions. This spread indicates the occurrence of permafrost in places where the regression result indicates slightly positive temperatures. The intention of the PGIM was rather to accept permafrost free areas within permafrost zone 1 than to include permafrost areas outside of zone 1. To include deviations towards lower temperatures in zone 1, the regression analysis was carried out twice. While all thermistors were used in the first iteration, only those thermistors with a measured MAGT below the modelled MAGT in the first iteration were used in the second iteration. Step 3 is summarized in Figure A (Supplementary Material).

**Step 4: mapping zone 1**

To map zone 1, the defined regression parameters a, b, and c were applied to a digital elevation and insolation model with 25 m resolution (DEM25 and DIM25, based on the swisstopo DHM25). Due to file size and computing limitations, we had to decrease the resolution of the gridded datasets compared to step 1. Beyond that, the DIM25 was produced in the same manner as in step 1. The temperature value of each 25 m raster cell of the PGIM was defined by:

(4) $\quad MAGT_{PGIM} = 19.497 \ + \ 4.532 \cdot 10^{-6} \cdot DIM25 \ - \ 0.008043 \cdot DEM25$

Depth-dependent 3D effects (Noetzli and Gruber, 2009), which were considered by the inverse distance weighting in our regression model, are not included in our map. In fact, such effects lose significance due to the lower resolution of the map, where insolation variations are spatially averaged within a 25 m raster cell. The temperatures on the map can therefore be interpreted as representing roughly the spatial average of mean annual ground temperatures in a cube with 25 m edge length: this corresponds to the horizontal extent of a raster cell and the typical depths of our reference data boreholes.

Zone 1 includes all areas with modelled negative ground temperatures and a buffer area with ground temperatures ranging between 0°C and 1°C. This buffer of 1 K corresponds to about the double standard error of our model output. The area of zone 1 with negative ground temperatures was labelled "Permafrost" and mapped in blue colours. The buffer area was mapped in yellow and is described as "possible patchy permafrost".

**2.2 Sensitivity analysis of the regression result**

The regression result depends on the following parameters: PISR, elevation and reference MAGT. Changes in these parameters will influence the regression result. Elevation is independent from external influences and therefore uncritical for the result. Reference MAGT can be influenced by environmental conditions as well as by measurement errors, which are not considered here. Our small to medium size statistical sample of measured ground temperatures might be distorted in comparison to the total statistical population. To test the sensitivity of our result to changes in the statistical sample, we carried out a 10-fold cross validation by randomly splitting the reference MAGT into 10 samples, nine of which were used as training data for our regression model and one as test data. The validation was carried out 10 times with each of the 10 samples as test sample, subsequently the resulting MAGT deviations of all 10 runs were averaged. The calculation of PISR values, especially in steep terrain, included several other parameters such as the distance threshold, a slope threshold, an aspect-dependent weighting factor and assumptions for the timing of snow coverage. Indeed, the model was optimized by applying these parameters. The PISR values are however not an independent statistical unit of a sample of observations but are all based on the same calculation. This means that they can introduce systematic errors to the model , e.g. due to simplified assumptions of the snow cover timing, but they are not the origin of random changes in the regression result.

**2.3 Mapping zone 2**

Zone 2 includes all forms of ice-rich permafrost such as rock glaciers or ice-rich talus slopes. Therefore, we defined areas in which the burial of ice or snow by rock fall can lead to the development of ground ice or in

which epigenetic ground ice may have been relocated due to ice creep. We carried out 9 work steps, as shown in Figure B (supplementary Material):

1. Avalanche snow and rock fall deposits were assumed to accumulate at the foot of slopes steeper than 40°. Potential locations of deposits were modelled by calculating runoff tracks from such slopes using ESRI ArcGIS with a 25 m DEM (Supplementary Figure B, a) This was done in areas above 2000 m a.s.l., as only few, azonal permafrost sites exist below in the Alps (e.g. Cremonese et al., 2011).

2. The runoff tracks were buffered by a 120 m wide belt as shown in supplementary Figure B, b. In their upper parts, the resulting areas correspond to the main tracks of snow avalanches and rock fall. Further downslope they represent potential rock glacier creep paths. The buffer was wide enough to include particularly broad rock glacier tongues.

These areas were then reduced stepwise by excluding spatial intersections with other datasets through:

3. Removal of all areas steeper than 30° (Supplementary Figure B, c), which barely contain ice-rich permafrost (Kenner and Magnusson, 2017). Snow avalanches seldom form deposits in such steep slopes and epigenetic segregation ice in talus slopes would creep downslope.

4. Removal of all vegetated areas (Supplementary Figure B, d) because they commonly consist of fine-grained soils at relatively low elevations, where ice-rich permafrost is generally absent in the European Alps (Hoelzle et al., 1993). Vegetation cover was deduced from orthophotos ("SWISSIMAGE" provided by swisstopo) using the SAVI Index (Huete, 1988). Vegetated/unvegetated areas within the resulting 25 m grid were homogenized by iteratively applying a classic 3x3 cell erosion and dilation operation.

5. Removal of maximal extents of Little Ice Age (LIA) glaciation (Supplementary Figure B, e), because glacier coverage is known to disrupt underlying permafrost (Reynard et al., 2003; Ribolini et al., 2010). This dataset was created by Maisch (1999).

6. Removal of lakes and glaciers (based on "swissTLM3D" provided by swisstopo) (Supplementary Figure B, e)

7. Removal of flood plains, which were defined as being areas with slope < 4° and intersected by rivers (based on "DHM25" and "swissTLM3D" provided by swisstopo).

8. The remaining polygons were then aggregated to fill small gaps, simplified and smoothed. After this, all areas listed above were again excluded from the reworked polygons (Supplementary Figure B, f) .

9. Zone 2 can overlap zone 1 and zone 1 was mapped with the higher priority, which implies that ice-rich permafrost can also occur within zone 1, where it is not distinguished from ice-poor permafrost.

In a final step, the resulting polygons were checked and manually edited if necessary. Some still contained areas in which surface bedrock excludes the development of ice-rich permafrost. In a few cases, parts of rock glaciers were missing due to errors in the reproduction of creep paths or due to small terrain steps steeper than 30°. Manual editing included two tasks: All areas showing a bedrock surface, infrastructure or > 50% vegetation cover (for some reason not captured by the SAVI index) were removed from zone 2. Missing parts of rock glaciers were added to zone 2 if at least parts of them were already captured by the automatic mapping approach. An exemplary editing task is shown in supplementary Figure C. The human polygon editor was not aware of the positions of the validation points during this process.

**2.4 Validation**

Using the same validation dataset, we validated the PGIM and two other permafrost maps of Switzerland in addition to compare the results: The Alpine Permafrost Index Map (APIM) created by Böckli et al. (2012) and the Potential Permafrost Distribution Map (PPDM) (Gruber et al., 2006), available online in the swisstopo web map service (swisstopo, 2018). A more detailed methodical background to the PPDM can be found in Haeberli (1975), Keller (1992) and Gruber et al. (2004). The permafrost maps were validated using a set of 98 evidence points of permafrost occurrence or absence, of which 10 represent the Swiss reference boreholes used to set up the regression model for PGIM zone 1 (Table 2). The reference boreholes are distinguished from the other records in the PGIM validation results. A more detailed verification, e.g. of modelled temperatures, was not possible due to the lack of data. The validation dataset partly consists of records collected by Cremonese et al. (2011), of which we only used direct evidence of permafrost occurrence or absence having exact coordinates. Further validation points were provided by continuous near ground surface temperature data (GST) measured at 38 automatic weather stations in the Swiss Intercantonal Measurement and Information System (IMIS) (Russi et al., 2003). To balance the number of permafrost and permafrost-free validation points, only IMIS stations above 2400 m a.s.l. were used, which mostly lie within the critical elevation belt of discontinuous permafrost. The IMIS stations measure near-surface ground temperature at 10 cm depth with a Campbell 107 temperature probe. Of these 38 IMIS stations, 33 register a constant zero curtain during winter and are therefore expected to be on permafrost-free ground (Hoelzle, 1992). The remaining 5 stations show quite constant winter GST between -3°C and -4°C and are located on active rock glaciers. They were therefore classified as permafrost sites. A few additional validation sites were added from different sources (Table 2).

All classes of the PGIM were attributed with the number of validation records lying within them indicating permafrost occurrence or permafrost absence. Additionally, zone 2 of the PGIM was validated against an inventory of 124 rock glaciers in the Albula Alps created by Kenner and Magnusson (2017).

**3 Results**

3.1 Linear regression analysis of MAGT

Predicting the ground temperatures of the ice-poor reference boreholes on the basis of elevation and PISR yields a correlation coefficient of 0.94 and a standard deviation of 0.58°C (Table 3, Figure 1). This highlights the strong dependency of ice-poor permafrost on these two factors and its relatively high predictability. Including ice-rich permafrost in this regression analysis causes a drastic drop of the correlation coefficient and thus in the predictability of permafrost (Table 3 and Figure 2).

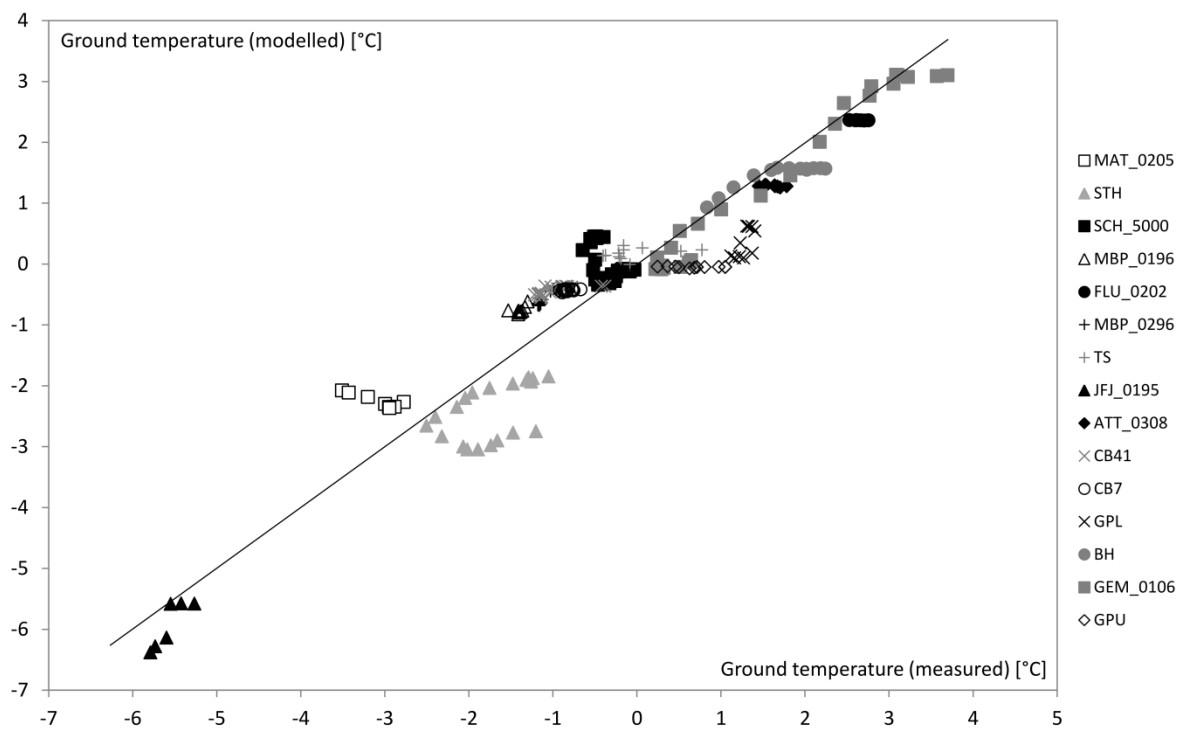

**Figure 1: Measured MAGT in 15 boreholes plotted against the modelled MAGT at the same locations. The regression line corresponds to formula (4) given in section 2.1. The borehole abbreviations are explained in table 1.**

Although thermistors of individual boreholes show clear deviations from the regression line, the cross-validation indicates a high robustness of the regression analysis result. The standard deviation between the modelled and measured ground temperatures stayed constant at 0.58 °C. The MAGT calculated during the cross validation differed from the MAGT calculated based on the entire set of reference temperatures by a mean value of -0.003° C and a standard deviation of 0.025 °C. The highest deviation found for a single thermistor was 0.14° C. Explanations for the deviations of single boreholes or thermistors are presented in chapter 4.1.

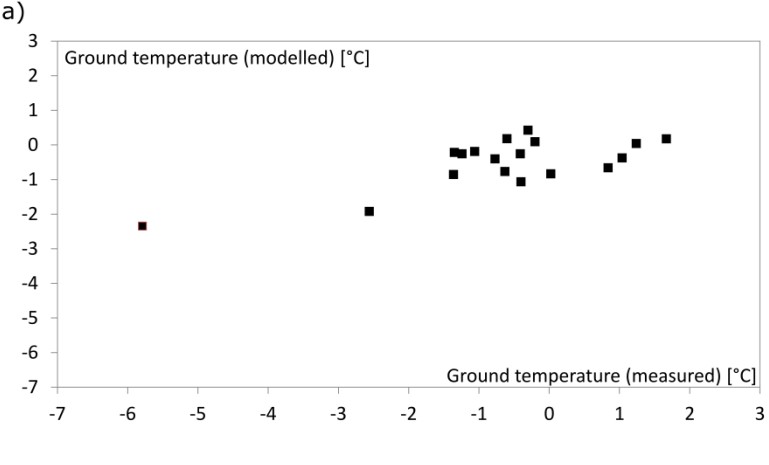

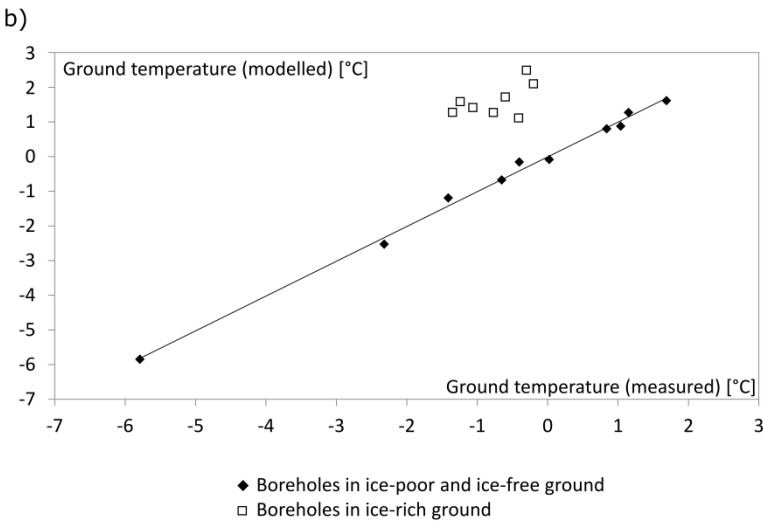

**Figure 2: Each data point represents a borehole and its measured and modelled mean annual ground temperatures at the depth with lowest temperatures. Included are the ice-poor boreholes 1-10 and all ice-rich boreholes in Table 2. The linear regression based on elevation and PISR shows no systematic relation between these two parameters and the ground temperatures when using both ice-poor and ice-rich boreholes for the regression (a), but a clear correlation appears when using only ice-poor or ice-free boreholes (b).**

3.2 Permafrost distribution in the PGIM

An example section of the PGIM is shown in Figure 3. The entire map is available online as a shapefile https://doi.org/10.5281/zenodo.1470165. Together, zones 1 and 2 indicate a potential permafrost area (area considered by the map to potentially contain permafrost) of 2000 km$^2$ in the Swiss Alps, which is considerably less than that indicated by the APIM (3710 km$^2$ (Böckli, 2013)) and also less than on the PPDM (2550 km$^2$ (Gruber et al. 2006)). To estimate the actual permafrost area (area effectively containing permafrost), Böckli (2013) considered all areas of the APIM with an index value > 0.5. This results in an area of 2160 km$^2$ for the APIM. The PGIM includes 830 km$^2$ in the core area of zone 1 and 600 km$^2$ in zone 2, of which maximum 90% are expected to include permafrost according to the validation output. This results in an actual permafrost area of < 1400 km$^2$ in the Swiss Alps, which corresponds to < 3.4% of the area of Switzerland. For comparison, Keller et al. (1998) gave a value of 4-6 %.

The permafrost distribution over elevation is shown in Figure 4 for different aspects. In very shady, north facing slopes, ice-poor permafrost occurs down to around 2550 m a.s.l. In south facing slopes, ice-poor permafrost terrain generally starts about 350 m higher. Ice-rich permafrost can occur in all aspects and has no sharp lower boundary. While the map indicates the highest frequency of ice-rich permafrost slightly above 2500 m a.s.l. for slopes facing northwest to northeast, it is at around 2600 m a.s.l. for slopes facing southeast to southwest.

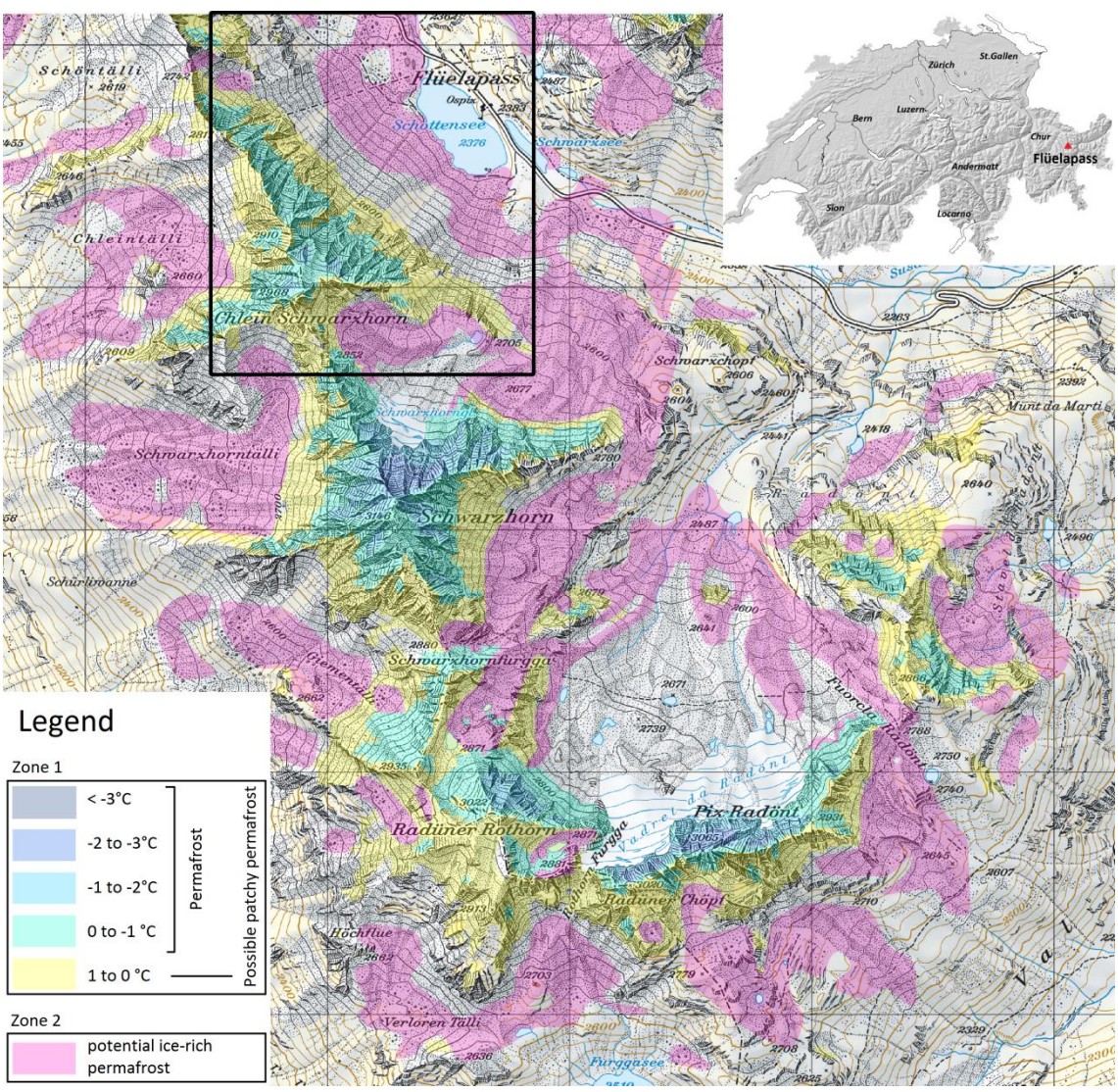

**Figure 3: Map section of the PGIM close to Flüelapass in the Eastern Swiss Alps (inset map of Switzerland), showing the permafrost distribution in two zones. The black frame is the sector shown in Figure 9. The map grid has a resolution of 1 km. (Map: pixmaps © (2017) swisstopo (5704 000 000))**

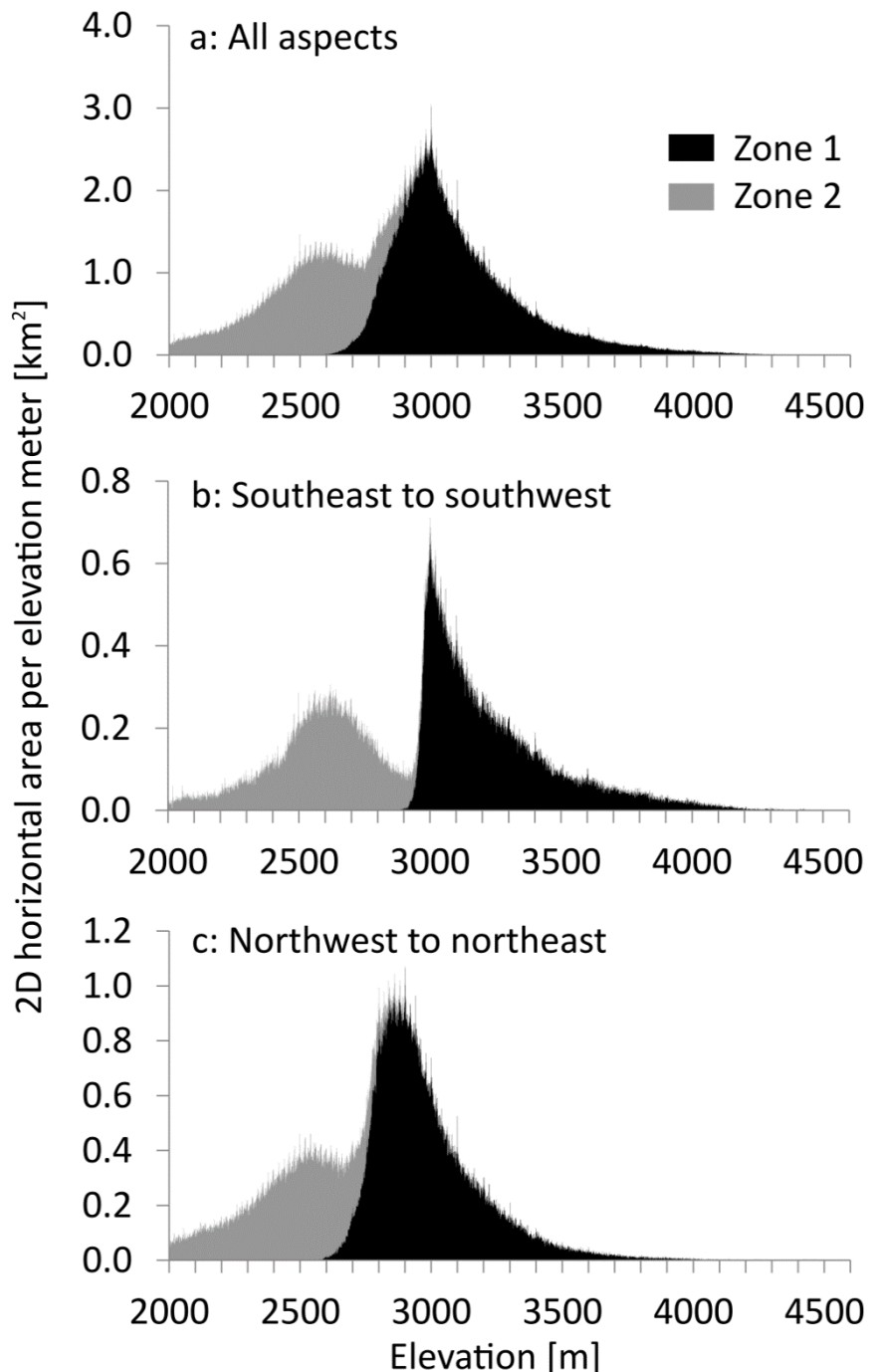

**Figure 4: Distribution of the PGIM zones 1 (only negative ground temperatures) and 2 over elevation. Part a shows the permafrost zonation over all aspects, part b for the aspects southeast to southwest and part c for aspects ranging between northwest and northeast. The permafrost gap appears between the two map zones, particularly in south facing slopes.**

3.3 Validation of the permafrost maps

The validation of the PGIM (Figure 5) confirms the high accuracy of zone 1. Only two validation sites representing ice-poor permafrost are located outside the core area of zone 1 labelled "permafrost" (Figure 5). In turn, no permafrost-free sites were located in the core area of zone 1. Zone 2 (potential ice-rich permafrost) includes 21 sites indicating permafrost and 2 indicating permafrost absence. Zone 2 furthermore includes 95.5%

of the rock glacier area registered in the Albula Alps inventory (Kenner and Magnusson, 2017). This value applies to the automatically created version of zone 2 before it was manually edited.

The validation of the APIM (Boeckli et al. 2012) is shown in Figure 6. The zones with a permafrost index of 0 (no permafrost) or 1 (definite permafrost) have a similar error rate as the corresponding classes in the PGIM, but contain less validation records. The indices between 0 and 1 contain a rather homogeneous ratio of permafrost and no-permafrost sites, an increase in permafrost frequency is only visible for the very highly indexed areas (> 0.8).

The validation result of the PPDM (Gruber et al. 2006) is shown in Figure 7. The different probability ranges reflect the actual permafrost frequency quite well for the high probability classes but show larger deviations for the lower classes. Several permafrost evidence points exist outside the permafrost zonation of this map.

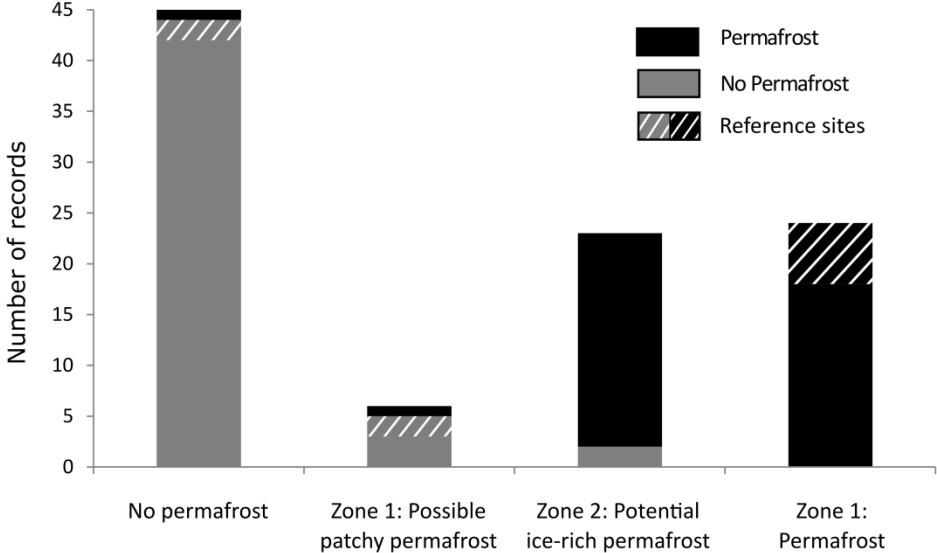

**Figure 5: Validation of the PGIM showing the number of validation points with permafrost occurrence and permafrost absence in each map class. The striped sites represent the boreholes used to set up the regression model for the PGIM.**

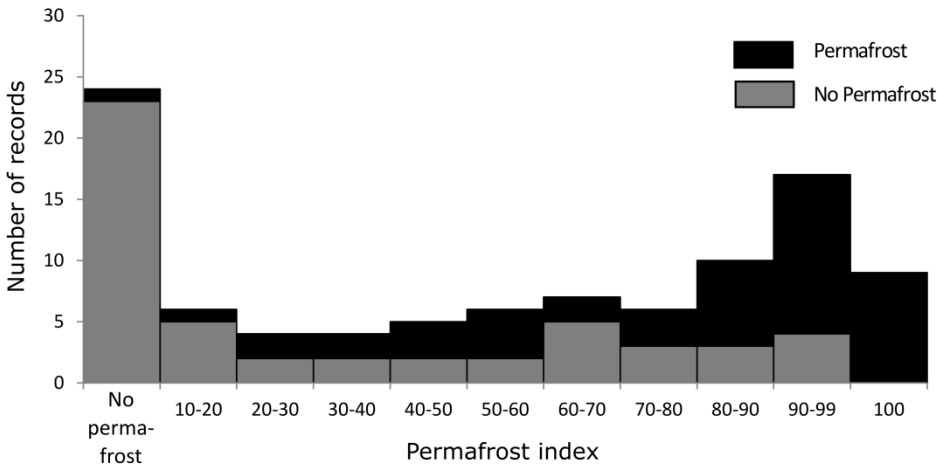

**Figure 6: Validation of the APIM (Boeckli et al. 2012) showing the number of sites with permafrost occurrence and permafrost absence for different permafrost probability ranges. As the map does not define classes but gives unique index values for each cell of the map, ranging from 0.1 to 1, these values were classified in 10 permafrost classes and a "No permafrost" class including all records outside the permafrost zonation.**

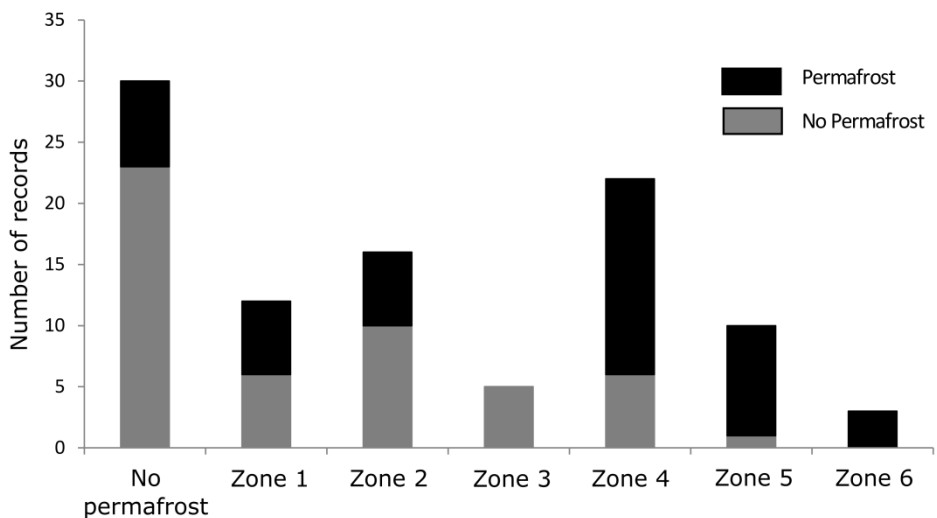

**Figure 7: Validation of the PPDM (Gruber et al. 2006) showing the number of sites with permafrost occurrence and permafrost absence in each map class. The zones were originally defined as follows: Zone 1 – local permafrost possible, patchy, discontinuous; Zone 2 - local permafrost possible, frequent patchy distribution; Zone 2 - local permafrost possible, patchy to extensive; Zone 4 – Extensive permafrost likely; Zone 5 – Extensive permafrost likely, increasing thickness; Zone 6 – Extensive permafrost likely, very thick in places, to over 100 m. The class "No Permafrost" includes all records outside the permafrost zonation.**

**4 Discussion**

**4.1 Permafrost predictability**

While the permafrost modelling based on the regression analysis was successful for ice-poor permafrost, it is not applicable for ice-rich permafrost (table 3). This makes ice-poor permafrost much better predictable than ice-rich permafrost. The high correlation coefficient achieved by the regression analysis is remarkable, because the borehole temperatures represent different landforms with strong differences in substrate and snow coverage. These factors, which influence ground temperatures (Haberkorn et al., 2015b; Hoelzle and Gruber, 2008; Schneider et al., 2012; Zhang, 2005), are represented in the regression result by rather small deviations of less than 1 K (Figure 8).

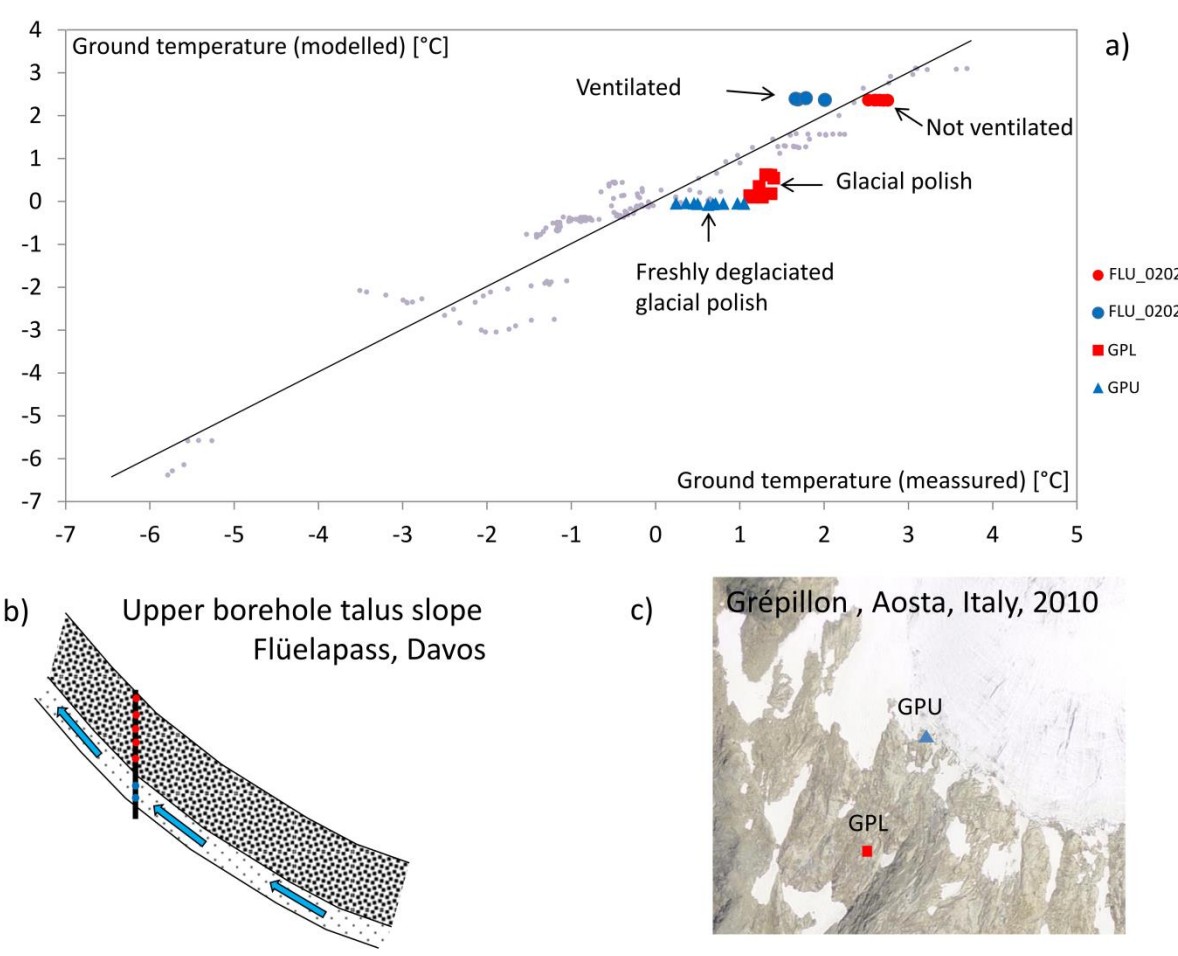

**Figure 8: The Flüela- (Eastern Swiss Alps) and Grépillon (Italian Alps) boreholes show examples of thermal disturbances. The lowermost 3 thermistors in Flüela (FLU_0202) are ventilated (Phillips et al., 2009) and thus deviate from the regression line. The Grépillon boreholes are drilled in a glacial polish, which can warm more efficiently than the talus surfaces at most of the other boreholes. The upper Grépillon borehole (GPU) was only recently deglaciated: whereas the uppermost thermistors have adapted to the new thermal conditions, there is a clear temperature gradient towards lower temperatures at greater depth. Here, the temperatures are still close to 0° C as a consequence of the former glaciation.**

Nevertheless, deviations exist due to advective cooling (Flüelapass; Figure 8a & 8b; (Phillips et al., 2009)), substrate characteristics (relatively warm glacial polish at the lower Grépillon borehole; Figure 8a & 8c) or

temperature disturbances due to former glaciation (upper Grépillon borehole; Figure 8a & 8c). Additional deviations might arise from the climate warming signal in the borehole temperatures. While near-surface temperatures might be in accordance with the current climatic conditions, temperatures at greater depth are still influenced by previous decades with colder climate conditions. As temperatures at several depths are included in our reference data set, depth-dependent deviations can occur. Our model for ice-poor permafrost does thus not represent a permafrost distribution which is in equilibrium with the current climate conditions but a snapshot of the current distribution of ice-poor permafrost, which is currently adapting to warmer climate conditions.

Ice-rich permafrost cannot be satisfactorily predicted based on surface energy fluxes and requires the consideration of mass wasting processes such as rock fall and avalanche activity, as well as creep rates and varying glaciation during the Holocene. As these processes are often not known in detail, the accuracy of the cartographic representation of ice-rich permafrost is limited, as discussed in section 4.3.

**4.2 Map uncertainty and accuracy**

The uncertainty of a map can be quantified by the validation points, which are clearly mapped as being permafrost or not. In the PGIM, definitive permafrost is indicated by the core area of zone 1. In the APIM definitive permafrost is indicated by a permafrost index of 1 (for validation, values higher than 0.994 were rounded to 1). The PPDM does not have a zone of definitive permafrost. Definitive permafrost absence is indicated on all three maps for areas outside the permafrost zonation. The PGIM could attribute 69% of the validation points to a definitive class, while the APIM reached 33% and the PPDM 23% (Figures 5-7).

Accuracy can be measured by the number of validation points wrongly attributed to a definitive class or by the plausibility of the description of a class. In the PPDM, 7 permafrost sites occur outside the permafrost zonation. The definitive permafrost classes of the APIM and the PGIM predict all validation points correctly - with the exception of one site (Emshorn-Oberems), which is wrongly attributed on both maps. A weakness of our accuracy analysis is that the landforms and geographical locations of the validation sites do not represent the natural variability. Terrain- or region related errors of the permafrost zones, which are not captured in this accuracy analysis are therefore possible.

The APIM includes almost all areas in Switzerland in which permafrost will occur and is therefore a useful tool to exclude permafrost at a certain location. However, similar to the PPDM it shows weaknesses in the reproduction of permafrost-free areas, while the PGIM performs better here. This might be caused by the 'elevational permafrost gap' phenomenon introduced in section 1. Figure 9a shows the example of the research site Flüelapass (Kenner et al., 2017), with a permafrost-free belt between the ice-poor and ice-rich zones.

Mapping solely based on thermal influences is not able to reproduce the permafrost gap and either neglects the ice-rich permafrost at the base of talus slopes (Figure 9b) or overestimates the permafrost further upslope (Figure 9b and 9c). This problem leads to a high number of permafrost-free validation points in the zones of medium permafrost probability on the comparison maps: for example in the 60-70 % probability zone on the APIM or the zone "local permafrost possible, patchy to extensive" on the PPDM (Figures 6 and 7). This may also cause the rather random distribution of permafrost-free validation points over the remaining probability classes of the APIM. In the PGIM the permafrost gap becomes visible when plotting the mapped permafrost area against

elevation as shown in Figure D (supplementary material). A more accurate identification of this permafrost gap is an important step because it enables a better planning of ice-sensitive infrastructure in alpine terrain.

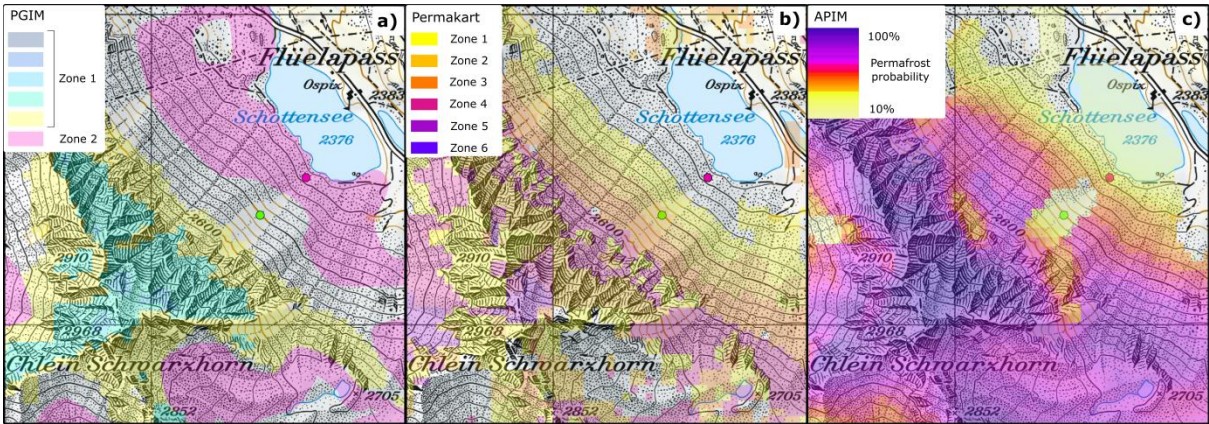

**Figure 9: Comparison of three permafrost maps at the research site Flüelapass (a: PGIM, b: PPDM (Gruber et al. 2006), c: APIM (Boeckli et al. 2012)). This example shows typical alpine permafrost distribution, with ice-rich permafrost at the base of a talus slope, a permafrost gap further upslope and permafrost in the rock wall above the talus slope. A borehole without permafrost (green dot (FLU_0202)) is located in the permfrost gap, another with ice-rich permafrost (pink dot (FLU_0102)) is at the base of the slope. (Map: pixmaps © (2017) swisstopo (5704 000 000))**

## 4.3 Challenges and possible future approaches in mapping ice-rich permafrost

Zone 2 of the PGIM has a relatively high uncertainty. The low number of permafrost-free validation points wrongly attributed to this zone (2 out of 33, see Figure 5) might rather overestimate the accuracy of the zone due to a general lack of permafrost-free validation points in talus slopes. However, there is very little ice-rich permafrost outside this zone, as indicated by the 95% representation of the Albula rock glacier inventory within the automatically created raw version of zone 2. Accordingly, zone 2 should not be interpreted as a reliable representation of ice-rich permafrost but rather as a best-guess including most of the ice-rich permafrost in Switzerland, with some bycatch of permafrost-free ground. The greatest challenges in mapping ice-rich permafrost are the correct representation of rock glaciers and the differentiation between loose rock sediments, which can contain ice-rich permafrost and bedrock, which cannot. Merging existing, manually created rock glacier inventories in Switzerland to a nationwide inventory would improve zone 2 as the model approach could be focussed on ice-rich talus slopes.

Kenner and Magnusson (2017) highlighted the influence of the combined effect of lithology and precipitation on the occurrence of ice-rich permafrost: Ice-rich permafrost is less frequent in sedimentary rock areas with high precipitation rates and relatively abundant in drier areas with crystalline or metamorphic lithology. These regional climate- and lithology induced differences are difficult to implement in a map and must be carefully interpreted by the user.

**4.4 Relevance of information on ground temperatures and ice content**

The PGIM is the first large scale permafrost map indicating permafrost temperature and ground ice content. The ice-rich permafrost in Zone 2, located in lower elevations than zone 1, typically has temperatures at or slightly below 0°C (PERMOS, 2016). Knowledge of the distribution of ice-rich and/or warm permafrost is particularly

important for engineering purposes, as ice affects the ground stability and bearing capacity strongest and should therefore be avoided during the infrastructure planning phase (Bommer et al., 2010). The construction of infrastructure on ice-rich permafrost can lead to destabilization of infrastructure through subsidence or creep induced by ice warming or even melting beneath the infrastructure. Construction activity and the subsequent use of infrastructure can lead to rapid changes in hydrology and ice content (Duvillard et al., 2019). The hydration

heat of concrete and heat from machinery in buildings are particularly problematic if the permafrost contains ice (Phillips et al., 2007). Permafrost in rock walls is very sensitive to climate fluctuations (Noetzli and Gruber, 2009) and rock temperatures influence rock slope instability (Davies et al., 2001; Gruber and Haeberli, 2007; Krautblatter et al., 2013). In general, substrates with negative ground temperatures require specially adapted construction materials to prolong the service-life of infrastructure (Bommer et al., 2008).

**4.5 Application to other regions**

The mapping approach of zone 1 can probably be adapted to other mountain regions or future climate scenarios without requiring any local ground temperature reference data. Formula 4 in section 2.1 defines the distribution of ice-poor permafrost solely by the parameters PISR and elevation as a proxy for MAAT. PISR can be calculated globally based on digital elevation models. By setting radiation to 0, Formula 4 represents a direct

conversion between elevation and MAAT. If the elevational MAAT distribution is known, elevation models for other climate regions can be adapted using this conversion and can then be used in the regression formula defined in this study. As for different climate regions, the elevation models can also be adapted to future scenarios of air temperatures. However, when adapting our results to a different climate, the transient effects have to be considered. As explained in 4.1., our model represents the actual current permafrost distribution in the

Swiss Alps and is therefore not in equilibrium with the current climate. This disequilibrium might be smaller or larger in future or in different world regions. The performance of our regression formula for permafrost temperatures in other world regions should be tested in a future study using the validation dataset of worldwide borehole temperatures. This validation dataset is currently in preparation (ESA, 2018). This universal application of our method would only be feasible for mapping ice-poor permafrost. Our approach for modelling ice-rich

permafrost can only be used for regions very similar to the Swiss Alps, as it is designed for non-arid, vegetated areas and requires special datasets such as information on past glaciation.

**5 Conclusions**

This study presents a new approach to map permafrost distribution in the Swiss Alps based on the differentiation

of ice-poor and ice-rich permafrost. The new approach highlights i) the high predictability of ice-poor, thermally induced permafrost based on a simplified surface energy balance and ii) the need for a different mapping

approach for ice-rich permafrost typically formed at the base of slopes by alpine mass wasting. This is important for mapping and local modelling, but also to develop scenarios of present, past and future permafrost evolution.

We conclude that:

- Using a simple linear regression analysis of solar radiation and elevation, ground temperature profiles of 15 boreholes in ice-poor or ice-free ground could be modelled with a clearly sub-Kelvin accuracy.

- The regression result that zone 1 of the map is based on can easily be adapted to different climate conditions: either spatially for different mountain regions in the world, or temporally for future climate scenarios in Switzerland.

- A major improvement has been achieved in defining permafrost-free areas (referred to as a permafrost gap in this study), which can be of particular interest for construction projects involving ice-sensitive infrastructure.

- The distribution of ice-rich permafrost outside the continuous zone of permafrost is better predicted by the analysis of mass wasting processes than by thermal influencing factors.

- The permafrost and ground ice map PGIM presented contributes towards an improvement in the accuracy of permafrost mapping in Switzerland.

- The two zones on the map provide clear information on their meaning (i.e. ground temperatures versus the potential occurrence of excess ice permafrost) rather than a probability value, and are thus easy to interpret.

While the distribution of ice-poor permafrost is predictable with a high accuracy, there is a relatively large uncertainty referring to ice-rich permafrost. To improve the mapping result here, a more detailed dataset on surface characteristics (talus vs. bedrock) and manually mapped rock glacier inventories are required. An improved data base is needed as well for the validation of permafrost maps in general. Currently available datasets are biased regarding aspect, elevation and landforms. In addition, evidence of permafrost absence in the belt of discontinuous permafrost is clearly lacking.

**Table 1: Reference boreholes provided by 1 - PERMOS (2016), 2 – WSL Institute for Snow and Avalanche Research SLF, 3 - Swiss Federal Office for the Environment FOEN, 4 - University of Lausanne, 5 - ARPA Valle d'Aosta. The uppermost 15 were used for the calculation of ground temperatures in zone 1 of the PGIM. The lowermost 8 were used to demonstrate the failure of this calculation if ice-rich and ice-poor boreholes are not distinguished (Table 3).**

| Line | Site name & provider | Abbreviation | Ground ice content | Elevation [m a.s.l.] | Longitude (WGS 84) | Latitude (WGS 84) | Time series |
|---|---|---|---|---|---|---|---|
| 1 | Breithorn [3] | BH | Ice-free | 2865 | 7.81785 | 46.14010 | 2016 – 2017 |
| 2 | Flüela 0202 [2] | FLU_0202 | Ice-free | 2501 | 9.94314 | 46.74687 | 2003 – 2005; 2009 |
| 3 | Tsaté [1] | TSA_0104 | Ice-poor | 3040 | 7.54844 | 46.10904 | 2009 – 2012; 2015 |
| 4 | Schilthorn 5200 [1] | SCH_5000 | Ice-poor | 2910 | 7.83442 | 46.55828 | 2006 – 2009; 2013 – 2015 |
| 5 | Stockhorn 6000 [1] | STo_6000 | Ice-poor | 3410 | 7.82419 | 45.98678 | 2011 – 2012; 2014 – 2016 |
| 6 | Les Attelas 3 [4] | ATT_0308 | Ice-free | 2741 | 7.27492 | 46.09659 | 2009 – 2010 |
| 7 | Jungfrau [1] | JFJ_0195 | Ice-poor | 3590 | 7.97316 | 46.54617 | 2010 – 2015 |
| 8 | Gemsstock [1] | GEM_0106 | Ice-free | 2940 | 8.61043 | 46.60125 | 2009 – 2010; 12; 15; 16 |
| 9 | Cima Bianchi 41 [5] (Italy) | CB41 | Ice-poor | 3094 | 45.91906 | 7.69249 | 2010 – 2011; 2014 – 2017 |
| 10 | Muot da Barba Peider 0196 [1] | MPB_0196 | Ice-poor | 2946 | 9.93109 | 46.49639 | 1997 – 2010; 2015 – 2016 |
| 11 | Muot da Barba Peider 0296 [1] | MPB_0296 | Ice-poor | 2942 | 9.93143 | 46.49657 | 2000 – 2011; 2015; 2016 |
| 12 | Cima Bianchi 7 [5] (Italy) | CB7 | Ice-poor | 3098 | 45.91920 | 7.69277 | 2010 – 2011; 2013 – 2017 |
| 13 | Grépillon, upper [5] (Italy) | GPU | Ice-free | 3047 | 7.05690 | 45.90990 | 2013 – 2017 |
| 14 | Grépillon, lower [5] (Italy) | GPL | Ice-free | 3000 | 7.05638 | 45.90919 | 2013 – 2017 |
| 15 | Matterhorn [1] | MAT_0205 | Ice-poor | 3288 | 7.67605 | 45.98232 | 2006 – 2007; 2009 – 2013 |
| 16 | Flüela 0102 [1] | FLU_0102 | Ice-rich | 2394 | 9.94516 | 46.74792 | 2005 – 2009; 2014 |
| 17 | Attelas 0108 [1] | ATT_0108 | Ice-rich | 2661 | 7.27307 | 46.09677 | 2009 – 2010; 12; 15; 16 |
| 18 | Attelas 0208 [1] | ATT_0208 | Ice-rich | 2689 | 7.27368 | 46.09674 | 2009 – 2010; 12; 15; 16 |
| 19 | Corvatsch 0200 [1] | COR_0287 | Ice-rich | 2672 | 9.82185 | 46.42878 | 2001; 2003 – 2008; 2010; 2011 2013 – 2017 |
| 20 | Lapires 1208 [1] | LAP_1108 | Ice-rich | 2500 | 7.28435 | 46.10611 | 2010; 2012; 2014 |
| 21 | Muragl 0299 [1] | MUR_0299 | Ice-rich | 2539 | 9.92735 | 46.50722 | 2010 – 2013; 2016; 2017 |
| 22 | Schafberg 0290 [1] | SBE_0190 | Ice-rich | 2754 | 9.92631 | 46.49737 | 2001 - 2016 |
| 23 | Ritigraben 0102 [1] | RIT_0102 | Ice-rich | 2690 | 7.84983 | 46.17469 | 2003; 2004; 2006; 2007; 2009; 2012; 2014; 2016 |

**Table 2: Validation sites and the zones assigned to them in the permafrost maps PGIM, APIM (Boeckli et al. 2012) and PPDM (Gruber et al. 2006). The bold typed sites at the bottom were used to set up the regression model for PGIM zone 1. Type: IMIS - IMIS station, BH - borehole, CS - construction site, RF - rock fall. Data providers: 1 – WSL Institute for Snow and Avalanche Research SLF, 2 - Cremonese et al. (2011), 3 - University of Lausanne, 4 – Swiss Federal Office for the Environment. 5 – University of Fribourg. Zones and probability classes of the maps: see Figures 5-7.**

| Type[provider] | Name | Permafrost | PGIM/ Temp. (mod) | APIM | PPDM | Elevation [m a.s.l.] | Longitude (WGS 84) | Latitude (WGS 84) |
|---|---|---|---|---|---|---|---|---|
| IMIS[1] | Boveire - Pointe de Toules | No | Zone 2 | 43 | Zone 4 | 2687 | 7.23722 | 45.98480 |
| BH[3] | Lapir2 | No | Zone 2 | 76 | Zone 2 | 2559 | 7.28345 | 46.10526 |
| IMIS[1] | Saas - Seetal | No | No perm. | No perm. | Zone 1 | 2477 | 7.87895 | 46.17137 |
| IMIS[1] | Trubelboden - Trubelboden | No | No perm. | No perm. | No perm. | 2459 | 7.58558 | 46.37096 |
| IMIS[1] | Lukmanier - Lai Verd | No | No perm. | 63 | No perm. | 2554 | 8.78352 | 46.60416 |
| IMIS[1] | Fully - Grand Cor | No | No perm. | 46 | No perm. | 2602 | 7.08964 | 46.19469 |
| IMIS[1] | Bernina - Puoz Bass | No | No perm. | 50 | No perm. | 2629 | 9.91588 | 46.44007 |
| IMIS[1] | Gandegg - Gandegg | No | No perm. | 72 | No perm. | 2710 | 7.76060 | 46.42926 |
| IMIS[1] | Kesch - Porta d'Es-cha | No | No perm. | 66 | Zone 1 | 2727 | 9.89813 | 46.62132 |
| IMIS[1] | Gornergrat - Gornergratsee | No | No perm. | 98 | Zone 5 | 2952 | 7.78359 | 45.98718 |
| BH[2] | Barthélemy les Rochers (Zinal) | No | No perm. | 35 | Zone 2 | 2519 | 7.59812 | 46.13660 |
| BH[2] | Neue Monte Rosa Hütte (Zermatt) | No | No perm. | 93 | Zone 1 | 2866 | 7.81233 | 45.95795 |
| IMIS[1] | Zermatt - Alp Hermetje | No | No perm. | No perm. | No perm. | 2409 | 7.70238 | 45.99799 |
| IMIS[1] | Goms - Treichbode | No | No perm. | No perm. | No perm. | 2428 | 8.22856 | 46.48912 |
| IMIS[1] | Julier - Vairana | No | No perm. | No perm. | Zone 1 | 2426 | 9.69231 | 46.47850 |
| IMIS[1] | Oberwald - Jostsee | No | No perm. | No perm. | No perm. | 2432 | 8.31595 | 46.54522 |
| IMIS[1] | Piz Martegnas - Colms da Prasonz | No | No perm. | No perm. | No perm. | 2429 | 9.53739 | 46.58009 |
| IMIS[1] | Bedretto - Cavanna | No | No perm. | No perm. | Zone 2 | 2420 | 8.51112 | 46.53268 |

| | | | | | | | | |
|---|---|---|---|---|---|---|---|---|
| IMIS[1] | Bernina - Motta Bianca | No | No perm. | No perm. | No perm. | 2447 | 10.02920 | 46.42057 |
| IMIS[1] | Davos - Hanengretji | No | No perm. | No perm. | No perm. | 2456 | 9.77400 | 46.78885 |
| IMIS[1] | Goms - Bodmerchumma | No | No perm. | 10 | Zone 2 | 2439 | 8.23251 | 46.42045 |
| IMIS[1] | Taminatal - Wildsee | No | No perm. | 59 | No perm. | 2468 | 9.39093 | 46.96836 |
| IMIS[1] | Eggishorn - Flesch | No | No perm. | No perm. | No perm. | 2500 | 8.09170 | 46.41680 |
| IMIS[1] | Bever - Valetta | No | No perm. | No perm. | No perm. | 2512 | 9.83713 | 46.53953 |
| IMIS[1] | Samnaun - Ravaischer Salaas | No | No perm. | No perm. | No perm. | 2512 | 10.33833 | 46.95637 |
| IMIS[1] | Weissfluhjoch | No | No perm. | 34 | No perm. | 2536 | 9.80911 | 46.82955 |
| IMIS[1] | Les Attelas - Lac des Vaux | No | No perm. | No perm. | No perm. | 2550 | 7.26988 | 46.10529 |
| IMIS[1] | Davos - Barentalli | No | No perm. | No perm. | Zone 2 | 2557 | 9.81941 | 46.69890 |
| IMIS[1] | Les Diablerets - Tsanfleuron | No | No perm. | 65 | No perm. | 2584 | 7.23939 | 46.31445 |
| IMIS[1] | Anniviers - Tracuit | No | No perm. | No perm. | No perm. | 2589 | 7.65639 | 46.12116 |
| IMIS[1] | Arolla - Breona | No | No perm. | No perm. | No perm. | 2602 | 7.56205 | 46.08742 |
| IMIS[1] | Anniviers - Orzival | No | No perm. | No perm. | Zone 4 | 2641 | 7.53536 | 46.18828 |
| IMIS[1] | Zermatt - Triftchumme | No | No perm. | 19 | Zone 4 | 2753 | 7.72738 | 46.04217 |
| CS[2] | Speichersee Totalpsee (Davos) | No | No perm. | 26 | Zone 2 | 2501 | 9.81109 | 46.83724 |
| CS[2] | Herrenabfahrt Corviglia (St. Moritz) | No | No perm. | 14 | Zone 2 | 2829 | 9.80023 | 46.50610 |
| BH[2] | Catogne (Bovernier) | No | No perm. | 21 | No perm. | 2331 | 7.10474 | 46.06012 |
| BH[2] | La Montagnetta (St. Jean/Grimentz) | No | No perm. | 0 | No perm. | 2270 | 7.55943 | 46.19472 |
| BH[2] | Barthélemy les Rochers (Zinal) | No | No perm. | 0 | Zone 2 | 2519 | 7.59812 | 46.13660 |
| BH[2] | Barthélemy les Rochers (Zinal) | No | No perm. | 0 | Zone 1 | 2519 | 7.59812 | 46.13660 |
| BH[2] | Emshorn (Oberems) | No | No perm. | 16 | Zone 1 | 2506 | 7.67602 | 46.26670 |
| BH[2] | Emshorn (Oberems) | No | No perm. | 0 | No perm. | 2506 | 7.67602 | 46.26670 |
| BH[2] | Felskinnbahn (Saas Fee) | No | No perm. | 68 | Zone 2 | 2585 | 7.91784 | 46.08137 |
| BH[2] | Illsee | No | No perm. | 0 | Zone 2 | 2359 | 7.63472 | 46.25945 |
| BH[2] | Lapires | No | No perm. | 97 | Zone 4 | 2650 | 7.28345 | 46.10526 |
| IMIS[1] | St. Niklaus - Oberer Stelligletscher | No | Zone 1: 0.4°C | 86 | Zone 2 | 2915 | 7.75054 | 46.16782 |
| BH[5] | Attelas 3 | No | Zone 1: 0.7°C | 69 | Zone 2 | 2741 | 7.27493 | 46.09660 |
| IMIS[1] | Arolla - Les Fontanesses | No | Zone 1: 0.9°C | 83 | Zone 4 | 2857 | 7.44542 | 46.02967 |
| IMIS[1] | Finhaut - L'Ecreuleuse | Yes | Zone 2 | 18 | No perm. | 2252 | 6.96409 | 46.10076 |
| IMIS[1] | Simplon - Wenghorn | Yes | Zone 2 | 46 | No perm. | 2424 | 8.04516 | 46.17802 |
| IMIS[1] | Piz Lagrev - Tscheppa | Yes | Zone 2 | 72 | Zone 1 | 2727 | 9.74488 | 46.45112 |
| IMIS[1] | Vinadi - Alpetta | Yes | Zone 2 | 82 | Zone 5 | 2729 | 10.44286 | 46.93178 |
| IMIS[1] | Saas - Schwarzmies | Yes | Zone 2 | 91 | Zone 5 | 2799 | 7.97436 | 46.12436 |
| CS[2] | Gruobtagfeld (Turtmanntal) | Yes | Zone 2 | 21 | No perm. | 2375 | 7.71797 | 46.20474 |
| CS[2] | Wasserscheide (Davos Parsenn) | Yes | Zone 2 | 56 | Zone 4 | 2620 | 9.80255 | 46.83391 |
| BH[2] | Gentianes | Yes | Zone 2 | 87 | Zone 5 | 2894 | 7.30226 | 46.08383 |
| BH[2] | Mont Dolin (Arolla) | Yes | Zone 2 | 49 | Zone 4 | 2597 | 7.46188 | 46.02634 |
| BH[2] | Mont Dolin, (Arolla) | Yes | Zone 2 | 30 | No perm. | 2574 | 7.46330 | 46.02634 |
| BH[2] | Ritigraben (Grächen) | Yes | Zone 2 | 51 | Zone 4 | 2639 | 7.84983 | 46.17470 |
| BH[2] | Seetalhorn (Grächen) | Yes | Zone 2 | 92 | Zone 5 | 2862 | 7.85911 | 46.17642 |
| BH[2] | Stafel-Seetalhorn (Grächen) | Yes | Zone 2 | 36 | Zone 4 | 2457 | 7.86022 | 46.18694 |
| BH[2] | Flüelapass (Davos) | Yes | Zone 2 | 29 | No perm. | 2500 | 9.94317 | 46.74688 |
| BH[2] | Lapires | Yes | Zone 2 | 61 | Zone 2 | 2505 | 7.28435 | 46.10612 |
| BH[2] | Schafberg I | Yes | Zone 2 | 74 | Zone 4 | 2752 | 9.92701 | 46.49655 |
| BH[2] | Schafberg II | Yes | Zone 2 | 61 | Zone 1 | 2729 | 9.92387 | 46.49909 |
| BH[2] | Murtèl-Corvatsch | Yes | Zone 2 | 83 | Zone 1 | 2666 | 9.82186 | 46.42879 |
| BH[2] | Muragl I | Yes | Zone 2 | 60 | Zone 4 | 2536 | 9.92784 | 46.50757 |
| BH[2] | Les Attelas1 | Yes | Zone 2 | 47 | Zone 4 | 2661 | 7.27308 | 46.09677 |
| BH[2] | Les Attelas2 | Yes | Zone 2 | 55 | Zone 4 | 2689 | 7.27369 | 46.09675 |
| BH[2] | Emshorn (Oberems) | Yes | No perm. | 0 | Zone 2 | 2506 | 7.67602 | 46.26670 |
| BH[2] | Muot da Barba Peider, lower shoulder | Yes | Zone 1: -0.1°C | 81 | Zone 4 | 2791 | 9.92891 | 46.49583 |
| RF[2] | Gemsstock (Andermatt) | Yes | Zone 1: -0.2°C | 99 | Zone 1 | 2911 | 8.61043 | 46.60125 |
| RF[2] | Chrachenhorn (Davos Monstein) | Yes | Zone 1: -0.4°C | 91 | Zone 5 | 2830 | 9.81226 | 46.68836 |
| BH[2] | Pointe du Tsaté | Yes | Zone 1: -0.4°C | 94 | Zone 5 | 3028 | 7.54696 | 46.10995 |
| BH[2] | Lagalp (Berninapass) | Yes | Zone 1: -0.4°C | 97 | Zone 2 | Restricted | Restricted | Restricted |
| RF[2] | Kärpf (Elm) | Yes | Zone 1: -0.6°C | 74 | Zone 4 | 2654 | 9.08917 | 46.91611 |
| CS[2] | Scex Rouge (Les Diablerets) | Yes | Zone 1: -0.6°C | 93 | No perm. | Restricted | Restricted | Restricted |
| CS[2] | Diavolezza (Berninapass) | Yes | Zone 1: -0.6°C | 98 | Zone 5 | 2993 | 9.96948 | 46.40975 |
| BH[2] | Schilthorn 51/98 | Yes | Zone 1: -0.7°C | 100 | Zone 4 | 2910 | 7.83462 | 46.55828 |
| CS[2] | Cabane des Vignettes (Arolla) | Yes | Zone 1: -0.9°C | 89 | Zone 1 | 3164 | 7.47555 | 45.98865 |
| CS[2] | Rothornhütte (Zermatt) | Yes | Zone 1: -0.9°C | 98 | Zone 4 | Restricted | Restricted | Restricted |
| CS[2] | Rifugio Camosci (Pizzo Cristallina) | Yes | Zone 1: -0.9°C | 94 | No perm. | 2903 | 8.53667 | 46.46444 |
| BH[2] | Arolla, Mt. Dolin | Yes | Zone 1: -1.0°C | 99 | Zone 5 | 2862 | 7.45473 | 46.02663 |
| BH[2] | Wisse Schijen (Randa) | Yes | Zone 1: -1.2°C | 89 | Zone 4 | 3039 | 7.74832 | 46.09635 |
| BH[2] | Stockhorn 61/00 | Yes | Zone 1: -2.7°C | 100 | Zone 4 | 3412 | 7.82420 | 45.98679 |
| CS[2] | Cabane Dent Blanche (Ferpècle) | Yes | Zone 1: -3.3°C | 100 | Zone 2 | Restricted | Restricted | Restricted |
| BH[2] | Jungfraujoch South | Yes | Zone 1: -3.9°C | 100 | Zone 2 | 3574 | 7.97306 | 46.54548 |
| BH[2] | Jungfraujoch North | Yes | Zone 1: -5.2°C | 100 | Zone 4 | 3602 | 7.97319 | 46.54611 |
| BH[2] | Eggishorn (Fiesch) | Yes | Zone 1: 0.6°C | 88 | Zone 1 | 2847 | 8.09365 | 46.42638 |
| **BH[2]** | **Flüelapass 0202** | **No** | **No perm.** | **18** | **Zone 2** | **2500** | **9.94317** | **46.74688** |
| **BH[2]** | **Gemsstock** | **No** | **Zone 1: 0.4°C** | **97** | **Zone 2** | **2940** | **8.61043** | **46.60125** |
| **BH[2]** | **Les Attelas 3** | **No** | **No perm.** | **73** | **Zone 4** | **2741** | **7.27492** | **46.09659** |
| **BH[4]** | **Breithorn** | **No** | **Zone 1: 0.7°C** | **81** | **Zone 2** | **2864** | **7.81785** | **46.14010** |
| **BH[2]** | **Muot da Barba Peider I** | **Yes** | **Zone 1: -1.0°C** | **99** | **Zone 6** | **2938** | **9.93092** | **46.49647** |
| **BH[2]** | **Tsate** | **Yes** | **Zone 1: -1.0°C** | **96** | **Zone 2** | **3040** | **7.54844** | **46.10904** |
| **BH[2]** | **Schildhorn 5200** | **Yes** | **Zone 1: -0.3°C** | **100** | **Zone 4** | **2910** | **7.83442** | **46.55828** |
| **BH[2]** | **Stockhorn 6000** | **Yes** | **Zone 1: -2.8°C** | **100** | **Zone 5** | **3410** | **7.82419** | **45.98678** |
| **BH[2]** | **Jungfrau** | **Yes** | **Zone 1: -5.3°C** | **100** | **Zone 6** | **3590** | **7.97316** | **46.54617** |
| **BH[2]** | **Hörnligrat (Matterhorn, Zermatt)** | **Yes** | **Zone 1: -2.0°C** | **100** | **Zone 6** | **3288** | **7.67605** | **45.98232** |

**Table 3: Results of the regression analysis on ground temperature in dependency of elevation and PISR. Left: Regression analysis used to map the PGIM. Centre: Regression analysis using only the 'coldest thermistor' in boreholes in homogeneous terrain (no ridges). Right: Same approach as in the central column but including the ice-poor boreholes shown in table 1.**

|  | Ice-poor permafrost (213 thermistors in 15 boreholes) | Ice-poor permafrost (coldest thermistor of 10 boreholes) | Ice-poor and ice-rich permafrost together (coldest thermistor of 10 ice-poor and 8 ice-rich boreholes) |
|---|---|---|---|
| Correlation coefficient | 0.944 | 0.998 | 0.523 |
| Standard error | 0.57° C | 0.16° C | 1.02° C |

## Acknowledgements

The authors sincerely thank all persons and institutions who supported this work by providing data. A major part of the reference ground temperatures was provided by the Swiss Permafrost Monitoring Network PERMOS. Paolo Pogliotti (ARPA Valle d'Aosta) and Christophe Lambiel (University of Lausanne) contributed valuable borehole temperature data to the study.  Ilja Burn is thanked for checking and manually editing the polygons representing zone 2 of the PGIM. Martin Schneebeli, the Editor Moritz Langer and two anonymous reviewers kindly provided constructive comments to the manuscript.

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

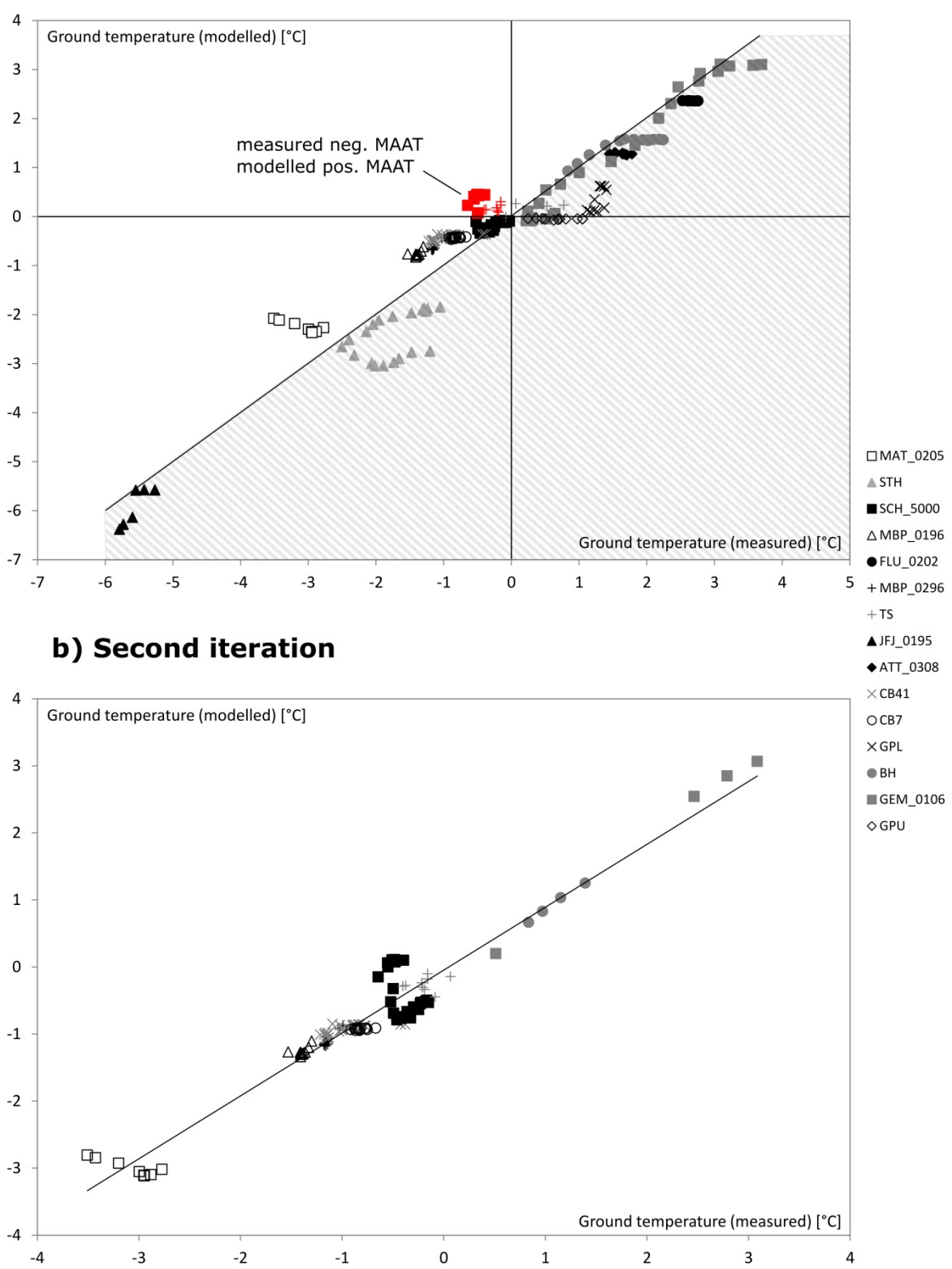

## a) First iteration

## b) Second iteration

**Figure A: The initial regression result (part a) showed positive and negative deviations of the modelled temperatures compared to the measured ones. This can lead to positive modelled temperatures while negative temperatures are actually present. Transferred into the map this can cause the indication of permafrost absence while permafrost is actually present. To avoid this, all temperature measurements which lie above the modelled temperatures (i.e. all data points in the shaded area below the regression line) were not used in the second iteration (part b). The regression in**

part b thus only includes temperature measurements which negatively deviate from the norm. This result was used to produce the map and ensures that the transition zone from permafrost to permafrost free terrain is also included in the permafrost zonation.

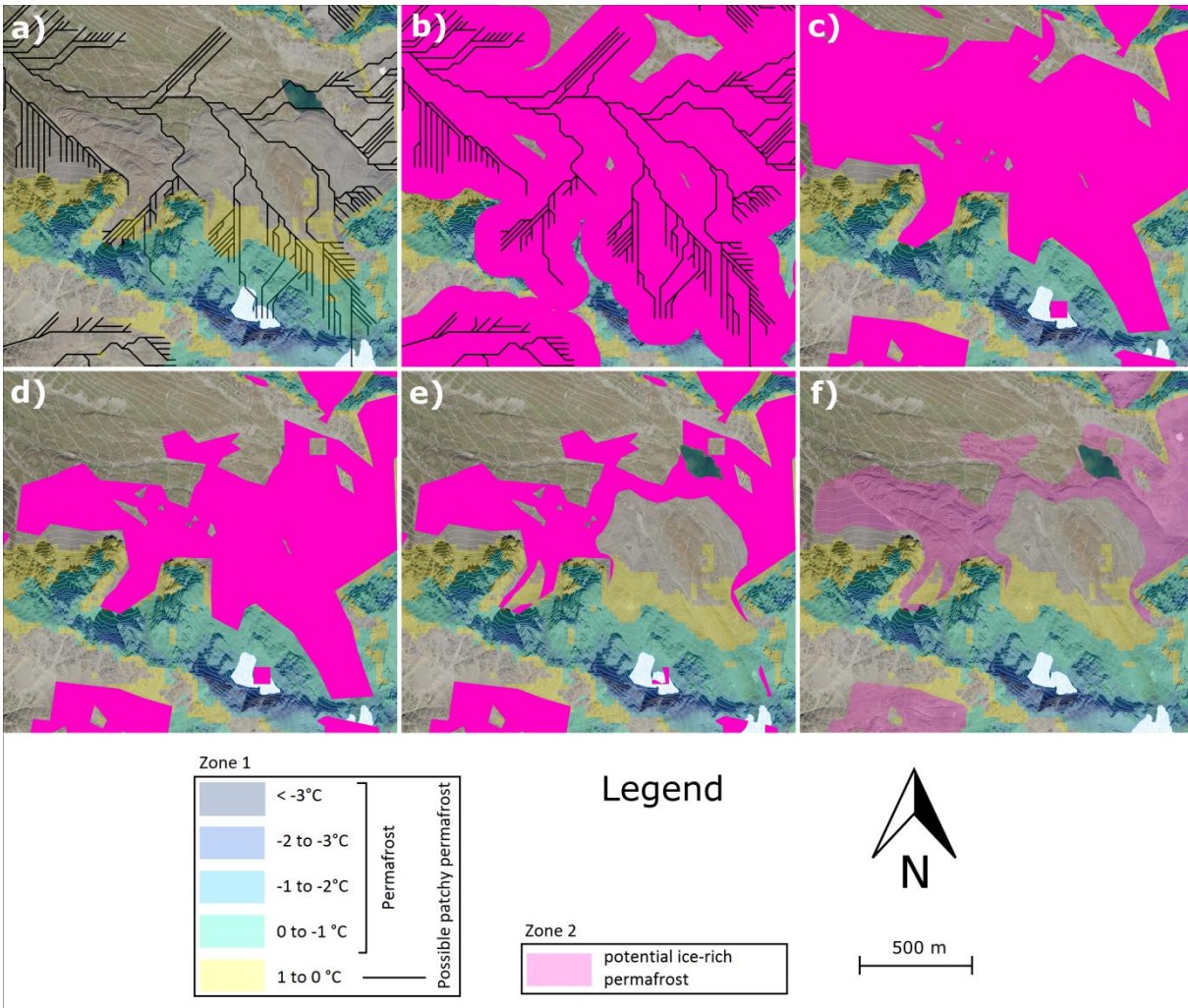

**Figure B: Parts a-f show the individual work steps described in section 2.3 to create zone 2 representing ice-rich permafrost. The example shows the area around rock glacier Muragl (Kenner, 2018; Maisch et al., 2003). In short: a) Step 1: runoff tracks; b) Step 2: Buffered runoff tracks; c) Step 3: erase areas steeper 30°; d) Step 4: erase vegetated areas; e) Step 5+6: erase LIA glaciation & Lakes; f) Step 8: Simplified and smoothed output map.**

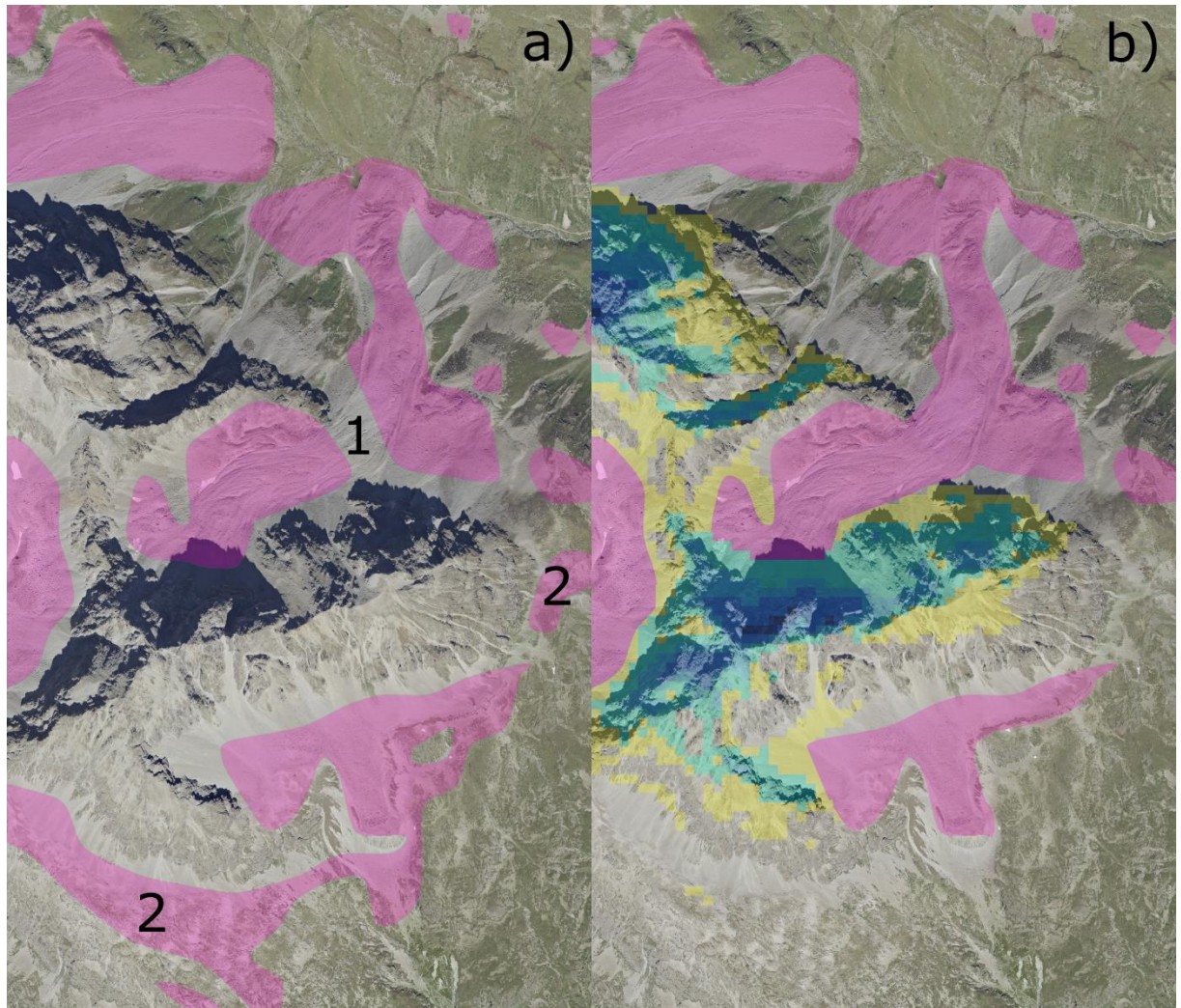

**Figure C: Part a shows the raw model output for zone 2. Part b shows the edited zone 2 together with zone 1. Index 1 shows a part of the rock glacier which was not captured by the model due to a terrain step steeper than 30° and included manually. Index 2 shows zones which were manually removed as they mainly include bedrock or vegetation-covered ground.**