# Peer review of "Distinguishing ice-rich and ice-poor permafrost to map ground temperatures and ground ice occurrence in the Swiss Alps"

_The Cryosphere, 2018_

## Referee Comment (RC1) · Anonymous Referee #1 · 11 Feb 2019

The manuscript presents a new country-wide permafrost map over Switzerland, which is based on statistical analysis of the national borehole-data network. During the mapping process, they distinguish between ice-poor and ice-rich permafrost, where the latter normally is associated to rock glaciers or talus accumulation, spatially often detached from the more continuous permafrost zone above in elevation. They used a multiple regressions approach to map MAGT for these two different zones.

The study is innovative as the permafrost types react differently on climate perturbations, and in alpine terrain ground ice has not been systematically mapped before. Thus, the study deserves attention. However, there are several issues which should be

resolved before considering publication.

In general, the paper is wordy, and can be shortened substantially. Tell the reader what you have done and avoid reporting-style. There were several parts I really did not follow even reading the passages several times. The following major points are identified:

1. Abstract. The abstract could be improved, what approach is chosen, what results are obtained which deserves attention?

2. Introduction: The introduction reads partly as a part of a discussion. An introduction should give the reader the background, not a discussion about what approach you have chosen. And it should end up in testable hypothesis, research questions or similar. This seems a bit mixed now, and should probably be re-formulated.

3. Methods: This chapter needs revision. To be honest, I still do not quite understand what the authors did in all details, and in which order. The description is full of report-style deviations in between substantial information. Sentences like "Attributing a MAGT to each thermistor is straightforward" have nothing to do in a scientific paper. Or what was "difficult" or not. I did not follow the handling of steeper slopes than 40 deg, in relation to PR calculations. Maybe a flow chart helps to describe the different steps. And: Give justification why you used different parameters, such like the selection of explanatory variables or why you consider points "5 times minimal distance of thermistors". Eq. 2: I understand you do a sort of interpolation, but I do not understand why and how you chose the factors. I do not follow p. 5, line 10 and following. The sensitivity tests: Nice, but now you mentioned more parameters included in the calculation of solar radiation?

Mapping of zone 2: Here you really need a flow chart, I cannot follow this, maybe some maps would help to illustrate the different steps, and a justification for e.g. the slope limits used, the size of the buffer zone around the runoff tracks etc. But as far as I could make out of it, you do an analysis identifying different types of mass wasting landforms (e.g. avalanche deposits, talus etc), do some manual editing and comparison to existing data sets, and then you do what to say that the landform is part

Interactive
comment
of zone 2? Were there new regression for these areas, or you used only the regression values from zone 1? I see you mentioned later that if including the boreholes in zone 2, the performance goes down. Ok, but is there some calculation to identify zone 2 permafrost or only the mapping? Please clarify these parts. It would help if you show maps of how you classified zone 2, this does not need to be in the main text, ok within an appendix if possible.

4. Validation: Ok, but could be part of the methods section, does not need an own chapter.

5. Results: Ok, show a map to present the results, this is a mapping exercise, and it is good for the reader to see a map then. The validation is good for zone 1, does not work on zone 2. I would prefer error matrix analysis instead of the histograms (or in addition) against real observations, which also would provide a classification performance measure in addition to the regression-R2s. I am not sure if the comparison to other modelling results (Böckli, Gruber) is a "validation". I think, no, validation is against something we know is true or false, the other maps are also models with their biases. You also show this again in the discussion (now with maps), the comparison should be a result.

6. Discussion: The first paragraph is more a result, as is the figure. Paragraph p.14, line 10 is repetition, the same first paragraph in 5.2. Please avoid redundancies in the manuscript. Finally, the discussion is very close to the Swiss conditions, and the comparison to earlier mapping approaches. A discussion should highlight and discuss the results to general science, which turns such a study from a technical report to a scientific contribution. Therefore, I miss comparisons to other areas, a discussion of the transferability of your approach, the evaluation of the use of other statistical approaches like GLMs or similar (see e.g. Hjort et al, recently in Nature Geoscience, or Aalto et al in GRL), the comparison of your model with such approaches.

7. Conclusions: The conclusions states in several places that your approach is a real

improvement, and this can be done in other regions. Probably, but then you need the same training data basis (which is exceptionally good in Switzerland), or are the regression coefficients universal? I do no think so. And if it so, why did you not test to transfer your approach, I guess it would be easy to transfer it at least into other areas in the Alps? What is the last conclusion point? I think you may consider reformulation of the conclusions.

In summary, the paper needs major improvements to convince the reader that the presented approach includes a major step forward. I am sort of convinced that this is important, but the paper is hard to read and follow. Especially I am a bit puzzled around how zone 2 is treated in the final product.

Distinguishing ground ice content is normal in Arctic permafrost regions, but little considered on mountain permafrost, and only related to clear landforms such like rock glaciers or frozen peat plateaus. This knowledge has be part of permafrost models, which is, as far as I see, the major message from this study.

---

## Referee Comment (RC2) · Anonymous Referee #2 · 22 Feb 2019

The paper presents a new mapping approach for mountain permafrost in Switzerland accounting for ground temperature and ice content. The study is based on regression analysis using borehole temperature collected in the Swiss Alps. The overall interest of this study is to propose a statistical approach to distinguish ice poor and ice rich permafrost in a mapping exercice, and to provide a more detailed and more accurate map of mountain permafrost distribution in Switzerland, representing permafrost # gaps # in its altitudinal distribution resulting of the combination of topoclimatic factors and ground ice content. The approach and objectives of the study are sounds and well suited for the journal, but it is very hard to provide a detailed and constructive review on the scientific content at the current stage. The writting misses dreadfully concise-

ness and precision. The paper can only be accepted after major revisions, notably rewritting of most sections to make it easier to understand and to follow the different steps. I have tried to formulate general comments to in order to guide the rewritting but I finally dropped many detailed comments as it was too messy. General : One strinking thing is the lack of references to the international research context : most references to previous work and knowledge focus on studies conducted in Switzerland, if not on the research team. Given that this study is submitted to an international journal of broad significance, one could expect that the international and broad significance of the paper is clearly stated and explained. Abstract : it lacks of precise results about the predicted permafrost distribution and the improvments achieved using this innovative approach. Stating that allowed a clear improvement is inappropriate and would first need to be desbribed. In general, the abstract is rather coarse. I suggest to rewrite it based on the following outlines (or similar, this is just a suggestion) : 1. State about the overall context, relevance and objective/research gaps and research questions of the study, 2. Briefly explain the chosen method, 3. Provide key results (as quantitatively as possible) 4. Explain the main implications of the results and main answer to research question.

Introduction : Rather badly organised also. The study is introduced as early as L15 with # The permafrost ground ice map. . .. #, but followed by decsription of the scientific background are given. This background is mainly based on studies from the 1st author and reference to the international context would be welcome. However, the lines dedicated to introducing the study (p3 L10-15) are very poor. Try to be more straight-forward and precise in your description : what is the specific approach you choose ? Why (based on the background you described above) ? What are the main expected results to fill the research gap ? This is roughly what the authors propose, but it is too general and the reader to not exactly get what will be presented in the study.

Methods : Here again, it is difficult to follow as the organisation of the method description is messy. Try to be more specific and more logical with the titles in order to ease

the reader following the approach. It starts with # mapping #, then with a # sensitivity analysis #, # a testing of zone 1 for zone 2 # while zone 2 is presented only afterwards. Similarly, the subsections are very unequal (2.3 is very short and one can wonder about its relevance). The short paragraph introducing Section 2 (p3 L 17-22) does not reflect the main outlines of the appraoch as it should, it just give general information about the maps. I wish I would have found a flow chart describing the methodological approach and this should be considered in the revised version. For example, illustrate the # buffer # (P3L25), the # mapped with blue colors # or # mapped in yellow # (L27). P5 L19 : I wonder if the regression coefficients would rather be in the results section while the method should rather describe the statistical approach. The meaning of the regression coefficients has to be briefly expanied (results). Your sensitivity analysis lacks of reference to common statistical methods. It is not clear whether this # bisected # sample is a common way to test model senstivity or if you have randomly decided it. More technically, I think that there is a misuse of # PGIM # in Eq. 3 since the acronym refers to the map and Eq. 3 is the regression analysis.

Section 3. Isn't it part of the methods ?

Section 4 and 5 are better written. However, in my sense, description of map features (general and more detailed at selected areas) is lacking as this study is a mapping exercice. What do your results show in term of permafrost distribution ? What is the elevation belt without permafrost for example ? Giving such information will, in my opinion, strongly broaden the significance of the results. Statistics given in 5.4 could be merge with such results (map) and therefore moved in section 4. They would be easier to get in a Table. Unless I missed something, the data that you use in Figure 1 and 2 are not very clear also : is it annual average ? multi-year average ? others ? which measurmeent years ? Title of 5.5 is not coherent with the content, even if it deals with ground temperature and ice content, the focus is more on implication of such a map for its use. Finally, as mentioned in the general comment, one expect that the authors place their study in an international context, at least in the discussion, and

this terribly missing.

Conclusions : they are poorly written. They are very general. They have to be written again with precise results and implications.

---

## Author Comment (AC1) · 22 Mar 2019

We sincerely thank both reviewers for their constructive feedback, which helped a lot to improve the paper. It became clear to us that the paper was not convincingly written and we carefully checked the entire manuscript trying to clarify and precise it.

Answers to reviewer 1

**The manuscript presents a new country-wide permafrost map over Switzerland, which is based on statistical analysis of the national borehole-data network. During the mapping process, they distinguish between ice-poor and ice-rich permafrost, where the latter normally is associated to rock glaciers or talus accumulation, spatially often detached from the more continuous permafrost zone above in elevation. They used a multiple regressions approach to map MAGT for these two different zones. The study is innovative as the permafrost types react differently on climate perturbations, and in alpine terrain ground ice has not been systematically mapped before. Thus, the study deserves attention. However, there are several issues which should be resolved before considering publication.**

**In general, the paper is wordy, and can be shortened substantially. Tell the reader what you have done and avoid reporting-style.**

Answer: We agree

Changes: We shortened the paper and deleted some repetitions and unsubstantial phrases

**There were several parts I really did not follow even reading the passages several times.**

Answer: Considering the following comments, I guess this refers mainly to the method section

Changes: The methods section was completely restructured and a supporting figure was added (Figure A) See detailed remark points below.

**1. Abstract. The abstract could be improved, what approach is chosen, what results are obtained which deserves attention?**

Answer: We agree

Changes: We rewrote the abstract following your suggestion

**2. Introduction: The introduction reads partly as a part of a discussion. An introduction should give the reader the background, not a discussion about what approach you have chosen. And it should end up in testable hypothesis, research questions or similar. This seems a bit mixed now, and should probably be re-formulated.**

Answer: We agree

Changes: We rewrote large parts of the introduction considering your remarks and removed the last discussion-like paragraph

**3. Methods: This chapter needs revision. To be honest, I still do not quite understand what the authors did in all details, and in which order.**

Answer: We realized that it was a major problem to follow our analyses.

Changes: We completely restructured the method section, switching to a chronological explanation of the work steps, which are now easier to follow as well as better justified.

**The description is full of report-style deviations in between substantial information. Sentences like "Attributing a MAGT to each thermistor is straightforward" have nothing to do in a scientific paper. Or what was "difficult" or not.**

Answer: You are right

Changes: We deleted phrases like that.

**I did not follow the handling of steeper slopes than 40 deg, in relation to PR calculations.**

Answer: This was not explained well.

Changes: We rewrote this and supported the understanding with two new formulae.

**Maybe a flow chart helps to describe the different steps.**

Answer: We renounced on a flow chart here but added a new formula.

Changes: We added a new formula.

**Give justification why you used different parameters, such like the selection of explanatory variables or why you consider points "5 times minimal distance of thermistors".**

Answer: The distance threshold was optimized empirically. How, is explained in more detail now.

Changes: We gave reasons for our choice of the explanatory variables.

**Eq.2: I understand you do a sort of interpolation, but I do not understand why and how you chose the factors. I do not follow p. 5, line 10 and following.**

Answer & Changes: This was completely rewritten explaining better what we did and why we did it.

**The sensitivity tests: Nice, but now you mentioned more parameters included in the calculation of solar radiation?**

Answer & Changes: This is now coherent with the first part of the method section

**Mapping of zone 2:  Here you really need a flow chart, I cannot follow this, maybe some maps would help to illustrate the different steps, and a justification for e.g.  the slope  limits  used,  the  size  of  the buffer  zone  around  the  runoff  tracks  etc.**

Answer: The slope limit was found in (Kenner and Magnusson, 2017), which is referred to here

Changes: We insert a figure with 6 maps that represent the different work steps.

**But as far as I could make out of it, you do an analysis identifying different types of mass wasting landforms (e.g.  avalanche deposits, talus etc.), do some manual editing and comparison to existing data sets, and then you do what to say that the landform is part of zone 2?**

Answer:  We defined a narrow as possible zone in which the development of ice-rich permafrost is possible. Whether permafrost actually exists at a certain location in this zone is unknown and impossible to model as it depends strongly on the formation history of the talus ground. (Was there coverage of avalanche snow or glacier ice? Was it buried fast and deep enough to be sufficiently protected from melting? Did large scale rock fall occur which created blocky and porous ground layers which enable cooling by ventilation effects?) All this is of course unknown for the large scale.

**Were there new regression for these areas, or you used only the regression values from zone 1?**

Answer: We used no regression for zone 2. This is now stated more clearly.

** I see you mentioned later that if including the boreholes in zone2, the performance goes down.  Ok, but is there some calculation to identify zone 2permafrost or only the mapping?  Please clarify these parts.  It would help if you show maps of how you classified zone 2, this does not need to be in the main text, ok within an appendix if possible.**

Answer: Only the mapping approach was used to identify zone 2.

Changes: As you suggested, we added a figure showing 6 maps which represent the different working steps for creating zone 2.

**4. Validation: Ok, but could be part of the methods section, does not need an own chapter.**

Answer: You are right.

Changes: We included the validation part in the methods

**5. Results: Ok, show a map to present the results, this is a mapping exercise, and it is good for the reader to see a map then.**

Answer: We agree

Changes: We moved figure 7 into the results section

**The validation is good for zone 1, does not work on zone 2. I would prefer error matrix analysis instead of the histograms (or in addition) against real observations, which also would provide a classification performance measure in addition to the regression-R2s.**

Answer: I am not sure if I understand this point. An elevation-dependent analysis of real data does not make so much sense as there is hardly data available for such an analysis and the existing data is biased in their distribution, depending on the motivation of their acquisition (an overrepresentation of north facing permafrost sites at elevations between 2500 and 3000 m asl.)

**I am not sure if the comparison to other modelling results (Böckli, Gruber) is a "validation". I think, no, validation is against something we know is true or false, the other maps are also models with their biases. You also show this again in the discussion (now with maps), the comparison should be a result.**

Answer: Well, this was never called a validation. We validated each of these maps using our set of validation records and the results were compared. It is a comparison of the performances of the different maps to show improvements achieved with the PGIM

**6. Discussion: The first paragraph is more a result, as is the figure.**

Answer & Changes: We slightly rewrote this part although we consider this paragraph as an interpretation of results

**Paragraph p.14,line 10 is repetition, the same first paragraph in 5.2.  Please avoid redundancies in the manuscript.**

Answer:  We agree.

Changes: We removed this.

**Finally, the discussion is very close to the Swiss conditions, and the comparison to earlier mapping approaches. A discussion should highlight and discuss the results to general science, which turns such a study from a technical report to a scientific contribution. Therefore, I miss comparisons to other areas, a discussion of the transferability of your approach, the evaluation of the use of other statistical approaches like GLMs or similar (see e.g. Hjort et al, recently in Nature Geoscience, or Aalto et al in GRL), the comparison of your model with such approaches.**

Answer & Changes: We added a section on the applicability of our approach to other regions and focused more on this issue in the entire paper. A comparison to lowland permafrost is however difficult as there are some considerable differences to (gravitational) processes taking place (or not taking place) in mountain permafrost due to the steep topography.

**7.  Conclusions: The conclusions states in several places that your approach is a real improvement, and this can be done in other regions.   Probably, but then you need the same training data basis (which is exceptionally good in Switzerland), or are the regression coefficients universal? I do not think so. And if it so, why did you not test to transfer your approach, I guess it would be easy to transfer it at least into other areas in the Alps?**

Answer & Changes: We added a section on this issue to the discussion.

**What is the last conclusion point? I think you may consider reformulation of the conclusions.**

Answer: We missed to discuss this point before the conclusions.

Changes:  We discuss the last conclusion point before and rewrote some of the conclusions.

**In summary, the paper needs major improvements to convince the reader that the presented approach includes a major step forward.  I am sort of convinced that this is important, but the paper is hard to read and follow. Especially I am a bit puzzled around how zone 2 is treated in the final product. Distinguishing ground ice content is normal in Arctic permafrost regions, but little considered on mountain permafrost, and only related to clear landforms such like rock glaciers or frozen peat plateaus.  This knowledge has be part of permafrost models, which is, as far as I see, the major message from this study.**

Answer: You are right: we rewrote the paper giving clearer explanations on our methods and clearer information on the relevance and improvement of the permafrost mapping in the international context.

Further changes: Some reference boreholes used to set up the PGIM zone 1 regression model were used together with the validation points to validate the permafrost maps APIM, PPDM and PGIM. They were accidentally part of the validation dataset. Using these sites for the PGIM is critically if validation and reference sites are not properly distinguished in the results. We reworked the figures showing the validation results and highlighted all reference sites in figure 4, which shows the validation of the PGIM. We furthermore highlighted all reference sites in table 2 and adapted the manuscript section.

**Answers to reviewer 2**

**The paper presents a new mapping approach for mountain permafrost in Switzerland accounting for ground temperature and ice content. The study is based on regression analysis using borehole temperature collected in the Swiss Alps. The overall interest of this study is to propose a statistical approach to distinguish ice poor and ice rich permafrost in a mapping exercise, and to provide a more detailed and more accurate map of mountain permafrost distribution in Switzerland, representing permafrost #gaps # in its altitudinal distribution resulting of the combination of topoclimatic factors and ground ice content. The approach and objectives of the study are sounds and well-suited for the journal, but it is very hard to provide a detailed and constructive review on the scientific content at the current stage. The writting misses dreadfully conciseness and precision. The paper can only be accepted after major revisions, notably rewritting of most sections to make it easier to understand and to follow the different steps.**

Answer: We agree to this general comment and made considerable changes to the paper to improve its clarity and preciseness.

**I have tried to formulate general comments to in order to guide the rewritting but I finally dropped many detailed comments as it was too messy. General : One strinking thing is the lack of references to the international research context : most references to previous work and knowledge focus on studies conducted in Switzerland, if not on the research team. Given that this study is submitted to an international journal of broad significance, one could expect that the international and broad significance of the paper is clearly stated and explained.**

Answer: We added more international references. They are listed at the end of this document. You will find some Swiss authors there as well, these studies are however focused on different world regions. Mapping of mountain permafrost is not carried out in many countries outside of the Alps (e.g., Norway, Iceland, Canada) and Switzerland plays a pioneer role in this field. This additionally explains the high

number of Swiss references. We better embedded the study in the international context by including a section on the applicability of the method to other regions.

Changes: More international references, new section on generic use of the method

**Abstract : it lacks of precise results about the predicted permafrost distribution and the improvments achieved using this innovative approach. Stating that allowed a clear improvement is inappropriate and would first need to be desbribed. In general, the abstract is rather coarse. I suggest to rewrite It based on the following outlines (or similar, this is just a suggestion) : 1. State about the overall context, relevance and objective/research gaps and research questions of the study, 2. Briefly explain the chosen method, 3. Provide key results (as quantitatively as possible) 4. Explain the main implications of the results and main answer to research question.**

Answer & changes: We rewrote the abstract based on your suggestions

**Introduction : Rather badly organised also. The study is introduced as early as L15with # The permafrost ground ice map.... #, but followed by decsription of the scientific background are given.**

Answer & changes: We rewrote also the introduction and reordered its structure

**This background is mainly based on studies from the 1st author and reference to the international context would be welcome.**

Answer & changes: We substituted the mentioned references by other authors

**However, the lines dedicated to introducing the study (p3 L10-15) are very poor. Try to be more straight-forward and precise in your description : what is the specific approach you choose ?Why (based on the background you described above) ? What are the main expected results to fill the research gap ?**

Answer: We rewrote that considering your suggestions.

Changes: The aims and approach of the study is now introduced at the end of the section.

**Methods : Here again, it is difficult to follow as the organisation of the method description is messy. Try to be more specific and more logical with the titles in order to ease the reader following the approach. It starts with # mapping #, then with a # sensitivity analysis #, # a testing of zone 1 for zone 2 # while zone 2 is presented only afterwards.**

Answer: You are right, methods were messy, see also comments to reviewer 1.

Changes: we rewrote them following a chronological order of the work steps and explained better what we did and why

**Similarly, the subsections are very unequal (2.3 is very short and one can wonder about its relevance). The short paragraph introducing Section 2 (p3 L 17-22) does not reflect the main outlines of the appraoch as it should, it just give general information about the maps. I wish I would have found a flow chart describing the methodological approach and this should be considered in the revised version. For example, illustrate the # buffer # (P3L25), the # mapped with blue colors # or # mapped in yellow #**

Answer: We agree with this remark

Changes: We restructured all this. A supporting figure was added visualizing the different steps leading to the delineation of zone 2.

**(L27).P5 L19 : I wonder if the regression coefficients would rather be in the results section while the method should rather describe the statistical approach. The meaning of the regression coefficients has to be briefly expanied (results).**

Answer:  The regression parameters have no special meaning, they are just technical values of the linear function describing the interaction of solar radiation and MAAT regarding their effect on ground temperatures. As these technical values give no further information to the reader than that what the reader can see in Figure 1 we have the feeling it is easier to give the values in the method section rather than starting an additional paragraph for that in the result section.

**Your sensitivity analysis lacks of reference to common statistical methods. It is not clear whether this # bisected# sample is a common way to test model senstivity or if you have randomly decided it.**

Answer: A common way to test the sensitivity of a model is to define critical input variables and to vary them based on a realistic assumption of their uncertainty. Bisecting the statistical population gives in our view a realistic indication of the uncertainty: The model result becomes more stable as more reference values are used to calibrate it. Stability grows however not linearly with the number of reference values but the stability decreases exponentially with decreasing reference values. This means doubling the number of reference values has a much smaller effect on model stability than bisecting the number of reference values. The bisecting method applied here will therefore rather underestimate model stability and can be considered as conservative estimate of the model sensitivity.

**More technically, I think that there is a misuse of # PGIM # in Eq. 3 since the acronym refers to the map and Eq. 3 is the regression analysis.**

Answer: Yes you are right

Changes: we changed "PGIM" to "MAGT(PGIM)"

**Section 3. Isn't it part of the methods?**

Answer: Yes it is.

Changes: We moved it to the methods

**Section 4 and 5 are better written. However, in my sense, description of map features (general and more detailed at selected areas) is lacking as this study is a mapping exercice. What do your results show in term of permafrost distribution ? What is the elevation belt without permafrost for example ? Giving such information will, in my opinion, strongly broaden the significance of the results.**

Answer: The permafrost-free belt has no fix elevation values but its upper boundary depends on aspect, slope and ground characteristic and its lower boundary on terrain form. We show a histogram in the supplementary material showing the permafrost distribution over elevation, including the permafrost gap.

**Statistics given in 5.4 could be merge with such results (map) and therefore moved in section 4. They would be easier to get in a Table.**

Answer: You are right.

Changes: We moved this to the results

**Unless I missed something, the data that you use in Figure1 and 2 are not very clear also : is it annual average ? multi-year average ? others? which measurmeent years ?**

Answer: It is a multiyear average.

Changes: We added the years in table 1.

**Title of 5.5 is not coherent with the content, even ifit deals with ground temperature and ice content, the focus is more on implication of such a map for its use.**

Answer & Changes: We renamed the title to: Practical relevance of information on ground temperatures and ice content

**Finally, as mentioned in the general comment, one expect that the authors place their study in an international context, at least in the discussion, and this terribly missing.**

Answer: You are right

Changes: As mentioned above we inserted a section, which discusses the applicability of the methods to other regions.

**Conclusions : they are poorly written. They are very general. They have to be written again with precise results and implications**

Answer & Changes: We partly rewrote the conclusions

Further changes: Some reference boreholes used to set up the PGIM zone 1 regression model were used together with the validation points to validate the permafrost maps APIM, PPDM and PGIM. They were accidentally part of the validation dataset. Using these sites for the PGIM is critically if validation and reference sites are not properly distinguished in the results. We reworked the figures showing the validation results and highlighted all reference sites in figure 4 which shows the validation of the PGIM. We furthermore highlighted all reference sites in table 2 and adapted the manuscript section .

International Literature

Azócar Sandoval, G., Brenning, A., and Bodin, X.: Permafrost Distribution Modeling in the Semi-Arid Chilean Andes, 877-890 pp., 2017.

Böckli, L., Brenning, A., Gruber, S., and Noetzli, J.: Permafrost distribution in the European Alps: calculation and evaluation of an index map and summary statistics, The Cryosphere, 6, 807-820, 10.5194/tc-6-807-2012, 2012.

Böckli, L.: Characterizing permafrost in the entire European Alps: spatial distribution and ice content, University of Zurich, Mathematisch-naturwissenschaftliche Fakultät., 2013.

Davies, M. C. R., Hamza, O., and Harris, C.: The effect of rise in mean annual temperature on the stability of rock slopes containing ice-filled discontinuities, Permafrost and Periglacial Processes, 12, 137-144, 2001.

Ebohon, B., and Schrott, L.: Modeling Mountain Permafrost Distribution. A new Permafrost Map of Austria, Ninth International Conference on Permafrost, Fairbanks, Alaska, 2008, 397 - 402,

ESA: Permafrost CCI Project: http://cci.esa.int/Permafrost, access: 04.03.2019, 2018.

Fiddes, J., Endrizzi, S., and Gruber, S.: Large-area land surface simulations in heterogeneous terrain driven by global data sets: application to mountain permafrost, The Cryosphere, 9, 411-426, 10.5194/tc-9-411-2015, 2015.

Gisnås, K., Etzelmüller, B., Lussana, C., Hjort, J., Sannel, A. B. K., Isaksen, K., Westermann, S., Kuhry, P., Christiansen, H. H., Frampton, A., and Åkerman, J.: Permafrost Map for Norway, Sweden and Finland, Permafrost and Periglacial Processes, 28, 359-378, doi:10.1002/ppp.1922, 2017.

Gruber, S.: Derivation and analysis of a high-resolution estimate of global permafrost zonation, The Cryosphere, 6, 221-233, 10.5194/tc-6-221-21012, 2012.

Hipp, T., Etzelmüller, B., Farbrot, H., Schuler, T. V., and Westermann, S.: Modelling borehole temperatures in Southern Norway – insights into permafrost dynamics during the 20th and 21st century, The Cryosphere, 6, 553-571, 10.5194/tc-6-553-2012, 2012.

Huete, A. R.: A soil-adjusted vegetation index (SAVI), Remote Sensing of Environment, 25, 295-309, 10.1016/0034-4257(88)90106-X, 1988.

Ishikawa, M.: Spatial mountain permafrost modelling in the Daisetsu Mountains, northern Japan, in: Permafrost, Eighth International Conference on Permafrost, Zurich, Switzerland, 2003, 020072388, 473-478,

Krautblatter, M.: Detection and Quantification of Permafrost Change in Alpine Rock Walls and Implications for Rock Instability, 2009.

Krautblatter, M., Funk, D., and Günzel, F. K.: Why permafrost rocks become unstable: a rock–ice-mechanical model in time and space, Earth Surface Processes and Landforms, 38, 876-887, 10.1002/esp.3374, 2013.

Magnin, F., Deline, P., Ravanel, L., Noetzli, J., and Pogliotti, P.: Thermal characteristics of permafrost in the steep alpine rock walls of the Aiguille du Midi (Mont Blanc Massif, 3842 m a.s.l), The Cryosphere, 9, 109-121, 10.5194/tc-9-109-2015, 2015.

Ribolini, A., Guglielmin, M., Fabre, D., Bodin, X., Marchisio, M., Sartini, S., Spagnolo, M., and Schoeneich, P.: The internal structure of rock glaciers and recently deglaciated slopes as revealed by geoelectrical tomography: insights on permafrost and recent glacial evolution in the Central and Western Alps (Italy–France), Quaternary Science Reviews, 29, 507-521, 10.1016/j.quascirev.2009.10.008, 2010.

Zhang, T.: Influence of the seasonal snow cover on the ground thermal regime: an overview, Reviews of Geophysics, 43, 1-23, 2005.

---

## Referee Report (RR1)

This second version of the paper ***Distinguishing ice-rich and ice-poor permafrost to map ground temperatures and ground ice occurrence in the Swiss Alps*** from *Kenner et al.,* has significantly improved in terms of text legibility. The permafrost mapping approach and/or the results' presentation has changed : in the first version the study presented 2 permafrost maps, while only 1 map with 2 zones is described in this second version, which clarifies the paper.

As already mentioned in the first review, it is evident that the authors focus on a main research gap which is to differentiate genesis processes of alpine permafrost by considering the role of mass-wasting processes and ice burrial, based on topography and surface characteristics, in addition to the well-known topoclimatic controlling factors. This study is therefore highly relevant for the mountain permafrost community and deserves publication in *The Cryopshere.*

Although this second version shows a clear improvement compared to the the first version, some paragraphs remain hard to read and understand. The results presentation and their discussion could be significantly improved. I therefore recommend this paper for publication after considering the following improvements.

**General comments**

**Abstract :** Try to group the sentences related to the background and those related to the methods to ease the reading. Highlight most relevant results, which are not only the evidence of a permafrost-free belt, but also outcome from the regression analysis and validation for example. Hint at the broad significance, not only « new information for users ».

**Methods :** I wonder about the relevance of presenting the « Mapping approach » before the regression analysis as it introduces concepts related to the regression (example of « the double standard error of our model output»). Some details are not necessary at this stage (example : « the buffer area was mapped in yellow ») and confuse the reader. At this stage of the method, it is in my opinion better to introduce the modelling approach rather than the mapping approach which is the final product and a way to express the model. Similarly, the regression approach is presented at the same time as the mapping approach (P4, L1-5) and it would be better to start from description L31 (P3) : explaining main predictors variables, then the regression analysis and finally the mapping approach.  This is a suggestion, but in the current state, the method section is still confusing.

In addition, there is one technical point which remains very unclear to me. In the methods (P4), it is stated that an aspect-dependent factor is used to account for long-wave solar radiation. Do the north faces receive such long-wave solar component ? Even though north faces have temperature close to the air temperature they remain warmer than the air. Why do the aspect-depent factor is 0 for North faces ? Wouldn't it be a way to account for this offset between air temperature and rock surface in shaded faces ?

In the step 1 you use a 2 m resolution DEM and in step 4 a 25 m DEM. Why not always using the best resolution DEM ? Can you explain your choices ?

Could you add a figure to illustrate Step 3 (regression results).

You propose a sensitivity analysis (P6) based on a « randomly bisected sample ». I wonder why not being aligned with former statistical studies and common statistical approach such as (Boeckli et al., 2012) using a 10-folds cross-validation . Wouldn't it provide more robust results ?

In your sensitivity analysis, you state that the PISR can not provoke random changes in the regression results because it is based on same calculation. However, you assume some snow-cover areas for 6

months, which might be very far from real-world situation with lower elevation getting snow free earlier than higher elevation. Considering the same « winter time » for all terrains might be a important limit in your approach and this might be explained and discussed.

**Results :** This section is messy. Please divide in sub-sections, presenting results of the regression analysis, sensitivity analysis and then the map. It is a pity that the pattern of permafrost distribution is not better described ut I understand that this is not the main message of the paper as reflected by the title. But giving a few statistics (min, max, mean elevation for example) for comparison with other maps would reinforce the results and would be interesting for the community. In my sense, it would strengthen the visibility of this paper.

**Discussion :** Section 4.4 is very poor. Either you develop a little bit more with examples showing how your map could support societal decisions and challenges, or you move these sentences in the introduction or perspective. In section 4.5, I am quite doubtful about the application of your mapping approach with future climate scenarios. Your suggestion doesn't account for transient effect. The same is true for your mapping approach and this might be introduced in the methods and discussed in the discussion. P17 L17-19 : this is not clear what should be tested (other areas ? future scenarios ?) and what is in preparation. Please clarify.

 **Detailed comment**

P2 L9 : « a ground temperature dataset »

P2 L20 : « to convert the energy balance results » sounds clumsy

P3 L14-15 : list the processes again, maybe in brackets, but the reader has lost track of the above description when reaching that point.

P3 : sometimes you use « Zone », and sometimes « zone » : be consistent.

P4 L15 : it is inaccurrate to state that most alpine ground surface are snow-covered during 6 months. I would suggest using « winter time », since snow cover duration is highly dependent on elevation and aspect.

P4 L17 : could you just explain why you chose a threshold of 40° ? Based on which assumption or background ?

P4 L23 : which « feedback » are you talking about ?

P4 L24 : I do not understand, you are describing processes of rock walls and speak about « wet avalanche » ? Are avalanche really a typical process of rock walls ? I think that most of snow accumulating on rock walls melts away, but doesn't accumulate over substantial thickness and surface area to trigger avalanche. Could you clarify ?

P6 L4 : remove « work » from « manner as in work step 1 ».

P7 L15-17 : what about permafrost forming in deglaciated rock walls (*e.g.* Wegmann et al., 1998) ?

P7 L32 : what do you mean ? « The human polygon editor » The authors ? Not aware of the position of the validation points ???

P8 L20 : what about talus slopes ? Are they considered ? Are some relevant data existing for validating the model ?
P15 L30-31 : it might be something missing in this sentence, not clear
P16 L15 : a « a » is missing in « permafrost »

P17 L5 : something wrong in the phrasing

---

## Author Response (AR2)

General comments

**Abstract : Try to group the sentences related to the background and those related to the methods to ease the reading. Highlight most relevant results, which are not only the evidence of a permafrost-free belt, but also outcome from the regression analysis and validation for example. Hint at the broad significance, not only « new information for users ».**

We reordered some sentences referring to background and methods and added further content on the outcome of the study as you suggested.

**Methods : I wonder about the relevance of presenting the « Mapping approach » before the regression analysis as it introduces concepts related to the regression (example of « the double standard error of our model output»). Some details are not necessary at this stage (example : « the buffer area was mapped in yellow ») and confuse the reader. At this stage of the method, it is in my opinion better to introduce the modeling approach rather than the mapping approach which is the final product and a way to express the model. Similarly, the regression approach is presented at the same time as the mapping approach (P4, L1-5) and it would be better to start from description L31 (P3) : explaining main predictors variables, then the regression analysis and finally the mapping approach. This is a suggestion, but in the current state, the method section is still confusing.**

You are completely right. This problem was easy to solve. We moved the entire paragraph which deals with description of the final map product : "It includes all areas with modeled negative ground temperatures….. possible patchy permafrost" from the beginning of section 2.1 to the very end of section 2.1 where the actual mapping of zone 1 is described in the subsection "step 4: mapping zone 1".

**In addition, there is one technical point which remains very unclear to me. In the methods (P4), it is stated that an aspect-dependent factor is used to account for long-wave solar radiation. Do the north faces receive such long-wave solar component ? Even though north faces have temperature close to the air temperature they remain warmer than the air. Why do the aspect-depend factor is 0 for North faces ? Wouldn't it be a way to account for this offset between air temperature and rock surface in shaded faces ?**

We do not fully agree that MAGT in north faces always stay warmer than the MAAT. I just recently read about the example from Kitzsteinhorn:

*"Mean annual crack top temperature(MACTT) was more than 1 °C lower (-3.3 °C) than MAAT in 2016."*

https://www.the-cryosphere-discuss.net/tc-2019-42/tc-2019-42.pdf

The examples given in Haberkorn et al. 2015 show rock temperatures under snow free conditions, which are during the summer months slightly higher than the air temperatures. These values originate however from a WNW facing slope where our factor is > 0. A slightly negative annual radiation balance in steep snow free or snow poor rock slopes facing exactly north is considered by us as well as possible as a slightly positive radiation balance.  Beyond that, the aspect dependent factor is of course just an approximation of the actual radiation balance in rock slopes.

**In the step 1 you use a 2 m resolution DEM and in step 4 a 25 m DEM. Why not always using the best resolution DEM ? Can you explain your choices ?**

Yes, this is simply because of the computing capacity and the size of the dataset. For small areas around the boreholes it is feasible to work with the high resolution DEM. When extrapolating our results to the entire Swiss Alps however, we had to switch to a lower resolution. We added a short explanation in the text.

**Could you add a figure to illustrate Step 3 (regression results).**

Yes, we added a figure to the supplementary material.

**You propose a sensitivity analysis (P6) based on a « randomly bisected sample ». I wonder why not being aligned with former statistical studies and common statistical approach such as (Boeckli et al., 2012) using a 10-folds cross-validation . Wouldn't it provide more robust results ?**

Well, what we proposed here was actually a 2-fold cross-validation. To access the robustness of the model, it seemed reasonable to shrink the training data by 50% instead of 10% as it is the case in a 10-fold cross validation. As you and the Editor were not happy with this approach, I changed to a 10-fold cross validation.

**In your sensitivity analysis, you state that the PISR cannot provoke random changes in the regression results because it is based on same calculation. However, you assume some snow-cover areas for 6months, which might be very far from real-world situation with lower elevation getting snow free earlier than higher elevation. Considering the same « winter time » for all terrains might be a important limit in your approach and this might be explained and discussed.**

Yes you are right here, although I would consider this as a systematic error of the model which has however little connection to the model's sensitivity. The latter rather represents a value for the stability of the model against (random) changes in the input data. We discussed this issue shortly in section 2.2

**Results : This section is messy. Please divide in sub-sections, presenting results of the regression analysis, sensitivity analysis and then the map. It is a pity that the pattern of permafrost distribution is not better described but I understand that this is not the main message of the paper as reflected by the title. But giving a few statistics (min, max, mean elevation for example) for comparison with other maps would reinforce the results and would be interesting for the community. In my sense, it would strengthen the visibility of this paper.**

We added 3 subsections:

3.1. Linear regression analysis of MAGT

3.2. Permafrost distribution in the PGIM

3.3 Validation of the permafrost maps

Furthermore we moved the histogram of permafrost distribution from the supplementary to the main manuscript. We gave some minimum elevations or ice-poor permafrost. Mean and max values are little meaningful in our opinion because they are controlled by topography (existing area per elevation zone).

**Discussion : Section 4.4 is very poor. Either you develop a little bit more with examples showing how your map could support societal decisions and challenges, or you move these sentences in the introduction or perspective.**

We have added information explaining why ice-rich substrates are problematic for the construction of mountain infrastructure and should thus be avoided during the planning of infrastructure. This underlines that a map indicating the presence of ice-rich permafrost is highly useful for engineers. Knowledge of permafrost temperatures is also useful in the choice of construction materials.

**In section 4.5, I am quite doubtful about the application of your mapping approach with future climate scenarios. Your suggestion doesn't account for transient effect. The same is true for your mapping approach and this might be introduced in the methods and discussed in the discussion.**

Yes you are right. We made supplements in two parts of the discussion:

4.1. "…Additional deviations might arise from the climate warming signal in the borehole temperatures. While surface near temperatures might be in accordance with the current climatic conditions, temperatures at greater depth are still influenced by previous decades with colder climate. As temperatures at several depth are included in our reference data set, depth dependent deviations can occur. Our model for ice-poor permafrost does thus not represent a permafrost distribution which is in equilibrium with the current climate conditions but a snapshot of the current distribution of ice-poor permafrost, which currently adapts to warmer climate conditions."

4.5. "…As for different climate regions, the elevation models can also be adapted to future scenarios of air temperatures. However, when adapting our results to a different climate, the transient effects have to be considered. As explained in 4.1. our model represents the actual current permafrost distribution in the Swiss Alps and is therefore not in equilibrium with the current climate. This disequilibrium might be smaller or larger in future or in different world regions."

**P17 L17-19 : this is not clear what should be tested (other areas ? future scenarios ?) and what is in preparation. Please clarify.**

Yes, thank you, this was ambiguously written. We rewrote it.

**Detailed comment**

**P2 L9 : « a ground temperature dataset »**

Thank you

**P2 L20 : « to convert the energy balance results » sounds clumsy**

Changed to: " to define a permafrost likelihood or index based on the energy balance results"

**P3 L14-15 : list the processes again, maybe in brackets, but the reader has lost track of the above description when reaching that point.**

We rewrote this sentence

**P3 : sometimes you use « Zone », and sometimes « zone » : be consistent.**

We adapted this.

**P4 L15 : it is inaccurate to state that most alpine ground surface are snow-covered during 6 months. I would suggest using « winter time », since snow cover duration is highly dependent on elevation and aspect.**

We changed to: during large parts of the year.

**P4 L17 : could you just explain why you chose a threshold of 40° ? Based on which assumption or background ?¨**

Yes we rewrote this section, gave a reference and some more explanations.

One problem is not discussed in the paper (hard to understand at this point of the paper and therefore confusing):  As we work with two different DEM resolutions for model calibration and map production (2m and 25m) slope as well calculated over different extents. In the map, 40° refers to an average slope over 75 m. (This corresponds to the edge length of 3 raster cells of the used DEM. Slope is calculated over 3x3 cells of the DEM.) This implies that in most cases, there are clearly steeper parts within this area from which the snow slides off and accumulates in gullies or less steep parts. Especially in southern slopes this causes a positive feedback of exposed rock -> higher albedo -> warming -> snow melt in less steep parts of the slope. See your following remark below.

**P4 L23 : which « feedback » are you talking about ?**

This is explained in the following sentence. We added a colon behind feedback and explained the positive feedback in more detail.

**P4 L24 : I do not understand, you are describing processes of rock walls and speak about « wet avalanche » ? Are avalanche really a typical process of rock walls ? I think that most of snow accumulating on rock walls melts away, but doesn't accumulate over substantial thickness and surface area to trigger avalanche. Could you clarify ?**

We refer to both, melt and avalanching. Avalanches includes here all types of sliding snow. This may be less often a typical slab avalanche but smaller sluffs of wet snow or small gliding avalanches. They regularly occur in steep south facing terrain during large parts of the year, including warm winter days.

**P6 L4 : remove « work » from « manner as in work step 1 ».**

Ok

**P7 L15-17 : what about permafrost forming in deglaciered rock walls (e.g. Wegmann et al., 1998) ?**

Here we are in the section describing the methodical approach for zone 2 (ice-rich permafrost). There are per definition no rock walls in zone 2.

**P7 L32 : what do you mean ? « The human polygon editor » The authors ? Not aware of the position of the validation points ???**

The polygons were edited by Ilja Burn (Acknowledgements) and the first author. The editing task took place on the background of an orthophoto. During editing the editors did not know where the validation points were located and had therefore no incentive for an subjective editing of zone 2 at the locations with validation points. (In deed only a very few validation points were affected by editing and the editing task was quite unambitious at these sites, i.e. correcting some rock glacier outlines)

**P8 L20 : what about talus slopes ? Are they considered ? Are some relevant data existing for validating the model ?**

Talus slope permafrost is considered by the 92 validation points of which several lie in talus slope. As discussed later, there is a bias towards permafrost occurrence in these validation points. Permafrost absence in talus slopes is unfortunately underrepresented.

**P15 L30-31 : it might be something missing in this sentence, not clear**

This was actually a supplement to the previous sentence. It was unclearly written we corrected this.

**P16 L15 : a « a » is missing in « permafrost »**

We could not find the missing a? In our manuscript it was there??

**P17 L5 : something wrong in the phrasing**

I guess a comma was missing.

Additional Changes: We adapted the regression coefficients in Formula (4). The values in the last manuscript version originated from an old version of the map and were forgotten to change. We apologize this mistake.

[revised manuscript text omitted]

**Supplementary material**

**a) First iteration**

[Figure]

Ground temperature (modelled) [°C]

measured neg. MAAT
modelled pos. MAAT

Ground temperature (measured) [°C]

Legend:
□ MAT_0205
▲ STH
■ SCH_5000
△ MBP_0196
● FLU_0202
+ MBP_0296
+ TS
▲ JFJ_0195
◆ ATT_0308
✕ CB41
○ CB7
✕ GPL
● BH
■ GEM_0106
◇ GPU

Zone 1

**Figure A**

**b) Second iteration**

Ground temperature (modelled) [°C]

Ground temperature (measured) [°C]

Figure A: The initial regression result (part a) showed positive and negative deviations of the modelled temperatures compared to the measured ones. This can lead to positive modelled temperatures while negative temperatures are actually present. Transferred into the map this can cause the indication of permafrost absence while permafrost is actually present. To avoid this, all temperature measurements which lie above the modelled temperatures (i.e. all data points in the shaded area below the regression line) were not used in the second iteration (part b). The regression in

**part b thus only includes temperature measurements which negatively deviate from the norm. This result was used to produce the map and ensures that the transition zone from permafrost to permafrost free terrain is also included in the permafrost zonation.**

[Figure]

**Figure B: Parts a-f show the individual work steps described in section 2.3 to create zone 2 representing ice-rich permafrost. The example shows the area around rock glacier Muragl (Kenner, 2018; Maisch et al., 2003). In short: a) Step 1: runoff tracks; b) Step 2: Buffered runoff tracks; c) Step 3: erase areas steeper 30°; d) Step 4: erase vegetated**

10  **areas; e) Step 5+6: erase LIA glaciation & Lakes; f) Step 8: Simplified and smoothed output map.**

[Figure]

**Figure C:** Part a shows the raw model output for zone 2. Part b shows the edited zone 2 together with zone 1. Index 1 shows a part of the rock glacier which was not captured by the model due to a terrain step steeper than 30° and included manually. Index 2 shows zones which were manually removed as they mainly include bedrock or vegetation-covered ground.

[Figure]

Figure C

**[3] nach oben:** : Distribution of the PGIM zones 1 (only negative ground temperatures) and 2 over elevation. Part a shows the permafrost zonation over all aspects, part b for the aspects southeast to southwest and part c for aspects ranging between northwest and northeast. The permafrost gap appears between the two map zones